# Cross-species comparison reveals therapeutic vulnerabilities halting glioblastoma progression

Leo Carl Foerster[1,2,13], Oguzhan Kaya[1,2,13], Valentin Wüst [2,3], Diana-Patricia Danciu [4], Vuslat Akcay [1,2], Milica Bekavac [1,2], Kevin Chris Ziegler [1], Nina Stinchcombe [1,2], Anna Tang[4], Susanne Kleber[1], Jocelyn L. Y. Tang[1], Jan Brunken[1,2], Irene Lois-Bermejo[1,2], Noelia Gesteira-Perez[1], Xiujian Ma[5], Ahmed Sadik [6], Phuong Uyen Le[7], Kevin Petrecca [7], Christiane A. Opitz [6], Haikun Liu[5], Christian Rainer Wirtz[8], Angela Goncalves [9,10,11], Anna Marciniak-Czochra [4,12], Simon Anders [3] & Ana Martin-Villalba [1] ✉

The growth of a tumor is tightly linked to the distribution of its cells along a continuum of activation states. Here, we systematically decode the activation state architecture (ASA) in a glioblastoma (GBM) patient cohort through comparison to adult murine neural stem cells. Modelling of these data forecasts how tumor cells organize to sustain growth and identifies the rate of activation as the main predictor of growth. Accordingly, patients with a higher quiescence fraction exhibit improved outcomes. Further, DNA methylation arrays enable ASA-related patient stratification. Comparison of healthy and malignant gene expression dynamics reveals dysregulation of the Wnt-antagonist SFRP1 at the quiescence to activation transition. SFRP1 over-expression renders GBM quiescent and increases the overall survival of tumor-bearing mice. Surprisingly, it does so through reprogramming the tumor's stem-like methylome into an astrocyte-like one. Our findings offer a framework for patient stratification with prognostic value, biomarker identification, and therapeutic avenues to halt GBM progression.

Glioblastoma (GBM) is the most frequent and lethal adult brain malignancy, with a median survival of only 15 months and limited therapeutic advances in the past decades[1]. GBM is particularly life-threatening because the rigid confines of the skull quickly lead to increased intracranial pressure and a high risk of herniation as the tumor grows. As in any cellular system, this growth is sustained by the organization of cells along a continuum of activation states. Like many cancers, GBM growth is fueled by plastic cancer stem cells (CSCs), which give rise to phenotypically diverse tumor cell populations[2,3]. Recent single-cell studies have successfully mapped these diverse transcriptomic profiles to neurodevelopmental cell types[4–10]. Yet, while

these neurodevelopmental states are highly plastic and able to regenerate each other[4], their transition dynamics are not fully understood.

In the healthy brain, diverse cellular states are generated by radial glia during development and by neural stem cells (NSCs) in adulthood[11]. While radial glia are mainly found in an active proliferative state during development, in mice, and likely in humans[12], v-SVZ NSCs maintain lifelong neurogenesis through dynamic transitions between distinct activation states[13]. Classically, these activation states include quiescent cells poised to divide, actively proliferating cells that drive immediate expansion, and differentiated cells with specialized

functions[14]. Functionally, transitions between these states maintain the NSCs' ability to react to external factors, for example, by transiently increasing the number of proliferating cells after brain injury[15] or through increasing the number of quiescent cells in aging[15,16]. Thus, different NSC activation states fulfill the organism's needs at various life stages[15,17–19]. GBM cells likewise occupy distinct activation states[10,20–22], and cycling CSCs in particular have been a focus of targeted therapies[5,6,23,24]. However, such approaches are complicated by the intrinsic plasticity of GBM cells, which replenish the CSC pool[22], leading to inevitable relapse.

There is a critical need to expand therapeutic options for GBM patients in order to achieve more durable treatment outcomes. Treatment with the alkylating agent temozolomide, for example, eliminates cycling cells but spares quiescent tumor cells, which eventually resume proliferation and lead to recurrence[25–27]. A similar effect is observed for adult murine NSCs, where temozolomide temporarily suppresses neurogenesis, only for quiescent NSCs to subsequently activate and replenish the stem cell pool[15]. These similarities highlight the value of comparing GBM to NSCs in order to better understand the mechanisms underlying activation state plasticity. Recent studies have shown, for instance, that astrocytes can become activated following injury[28,29], transitioning from a dormant (i.e., long-term quiescent) to an NSC-like state by switching from a characteristic astrocytic to an NSC methylome[30]. While GBM methylation has been profiled at single-cell resolution[21,31], it remains unclear whether GBM cells mimic the methylation remodeling found in healthy v-SVZ populations. To address these questions, we investigate Activation State Architectures (ASAs): unique structural features that represent the organized distribution of activation states in each tumor. These ASAs have the potential to serve as a prognostic marker, support patient stratification, and inform personalized therapeutic interventions.

In this work, we decode individual patient ASAs by aligning single GBM cell transcriptomes within a reference NSC ASA from the adult murine brain using a dedicated tool, ptalign. These unique ASAs are integrated into mathematical models to explain GBM population dynamics and identify parameters influencing tumor growth. Using ptalign, we compare inter-tumor gene expression dynamics in 51 GBMs to the healthy counterpart, identifying the secreted Wnt antagonist SFRP1 as recurrently dysregulated. Reintroducing SFRP1 stalls tumor growth in vivo and reprograms the tumor from an NSC-like into an astrocyte-like methylome, significantly improving the overall survival of tumor-bearing mice. Our approach establishes a framework for the inference of comparative tumor hierarchies, enabling patient stratification and target discovery.

## Results

### ASAs in the fetal and adult mammalian brain

During brain development, neural progenitor cells are committed to generating differentiated progeny and are eliminated upon completion of their developmental role. In contrast, adult somatic stem cells are maintained to ensure lifelong tissue homeostasis through continuous progeny generation, analogous to CSCs, which persist to support ongoing tumor growth. To enable a reference-based prediction of GBM ASA, we leverage their similarity to healthy NSCs. Ideally, we would reference adult human NSC lineages, as these serve as a model for how tumor cells might organize and differentiate within a similar environment. However, datasets capturing adult human neurogenesis remain largely unavailable[32–35]. Instead, we turn to the adult mouse v-SVZ, a region that harbors quiescent NSCs which populate a lineage of quiescent and active states and ultimately give rise to different neural cell types[11,15,36] (Fig. 1a). scRNA-seq provides a snapshot of the v-SVZ NSC pool, allowing ASA and state transitions to be decoded by pseudotime analysis. We compiled a 14,793-cell murine v-SVZ NSC lineage dataset[15,19,37] and fit a NSC-to-neuron differentiation trajectory by diffusion pseudotime (Supplementary Fig. 1a, b). In this trajectory,

we delineated distinct activation states including quiescence (Q), activation (A), and differentiation (D); collectively referred to as QAD (Fig. 1b and Supplementary Fig. 1c). This approach places v-SVZ astrocytes at the lineage onset, supported by their transcriptional similarity to NSCs and their latent neurogenic potential[30]. The QAD stages in the NSC ASA are identifiable from pseudotime alone, and we derived a 242-gene pseudotime-predictive gene set (SVZ-QAD: "Methods"; Supplementary Fig. 1d and Supplementary Data 1) to facilitate their comparison across species.

To contrast ASAs in fetal and adult neurogenesis against GBM and include the healthy human context, we examined over 600,000 single-cell transcriptomes from human cortical development[38]. These cells exhibited a similar lineage progression to adult NSCs, and mapping QAD-stage frequencies over developmental time revealed a low prevalence of quiescent cells that increased in late development but did not surpass 5% (Fig. 1c and Supplementary Fig. 1e). A similar lineage and low incidence of quiescent cells were found in human brain organoids (HBOs), which mimic cortical development[39] (Supplementary Fig. 1f–h). Thus, although fetal and adult neurogenesis share a similar progression, the adult brain ASA is characterized by a significantly increased abundance of Q-stage cells.

We further analyzed QAD-stages in 399 bulk GBM samples from TCGA[40] and Wu et al.[24]. Patients in this cohort had an average age at diagnosis of 59 (Fig. 1d), underscoring the late emergence of these tumors within the adult brain[1]. We found Q-stage gene expression enriched among GBMs (Fig. 1e), consistent with the increased abundance of quiescence in the adult NSC ASA. Single-cell profiling of a GBM patient derived orthotopic xenograft (PDX, pseudonymously T6) supported this finding, identifying a tumor ASA with substantial Q-stage expression (Fig. 1f) within a UMAP structure closely resembling the adult v-SVZ NSC lineage (Fig. 1b). We identified distinct cell morphologies in this tumor (Fig. 1f), which, along with other features of the T6 GBM, were recapitulated in cells implanted into the fetal context of HBOs[41,42] (henceforth as patient derived allografts, PDAs; Supplementary Fig. 2). Finally, comparing QAD-stage frequencies in the T6 PDX with young and old adult v-SVZ populations and late cortical development revealed the greatest similarity in ASA between GBM and the aged v-SVZ NSC lineage (Fig. 1g). We conclude that referencing fetal development to decode the ASA of GBM may fail to capture relevant aspects of tumor biology due to the lack of a quiescent state. Instead, adult NSC lineages provide a more accurate framework, motivating a comparative view of GBM organization grounded in an adult v-SVZ NSC QAD-ASA.

### GBM ASAs inferred by ptalign

Having established parallels between GBMs and the adult v-SVZ NSC lineage, we sought to systematically resolve distinct patient ASAs through the lens of healthy lineage dynamics in an approach we termed pseudotime alignment, or ptalign. This tool maps tumor cells onto a reference lineage trajectory (here the murine v-SVZ NSC lineage), thereby resolving both individual cell stages and the transitions between them (Fig. 2a). A pseudotime-similarity metric derived from gene expression correlations between a query cell and regularly sampled increments along the reference pseudotime underlies this mapping (Fig. 2b). Because cellular differentiation is governed by gradual transcriptional changes, each cell's pseudotime-similarity profile serves as a signature of its position in pseudotime. By normalizing these profiles and focusing on their shape, a neural network is able to learn an efficient mapping between cellular similarity profiles and pseudotimes (Fig. 2c). For training, we use similarity profiles of the pseudotime-masked reference, which serve as a convenient ground truth. Cycling cells are excluded to limit inference to non-branching pseudotimes. The trained network then predicts 'aligned' pseudotimes for query (tumor) cells from their pseudotime-similarity profiles (Fig. 2d), applying additional heuristics to exclude out-of-distribution

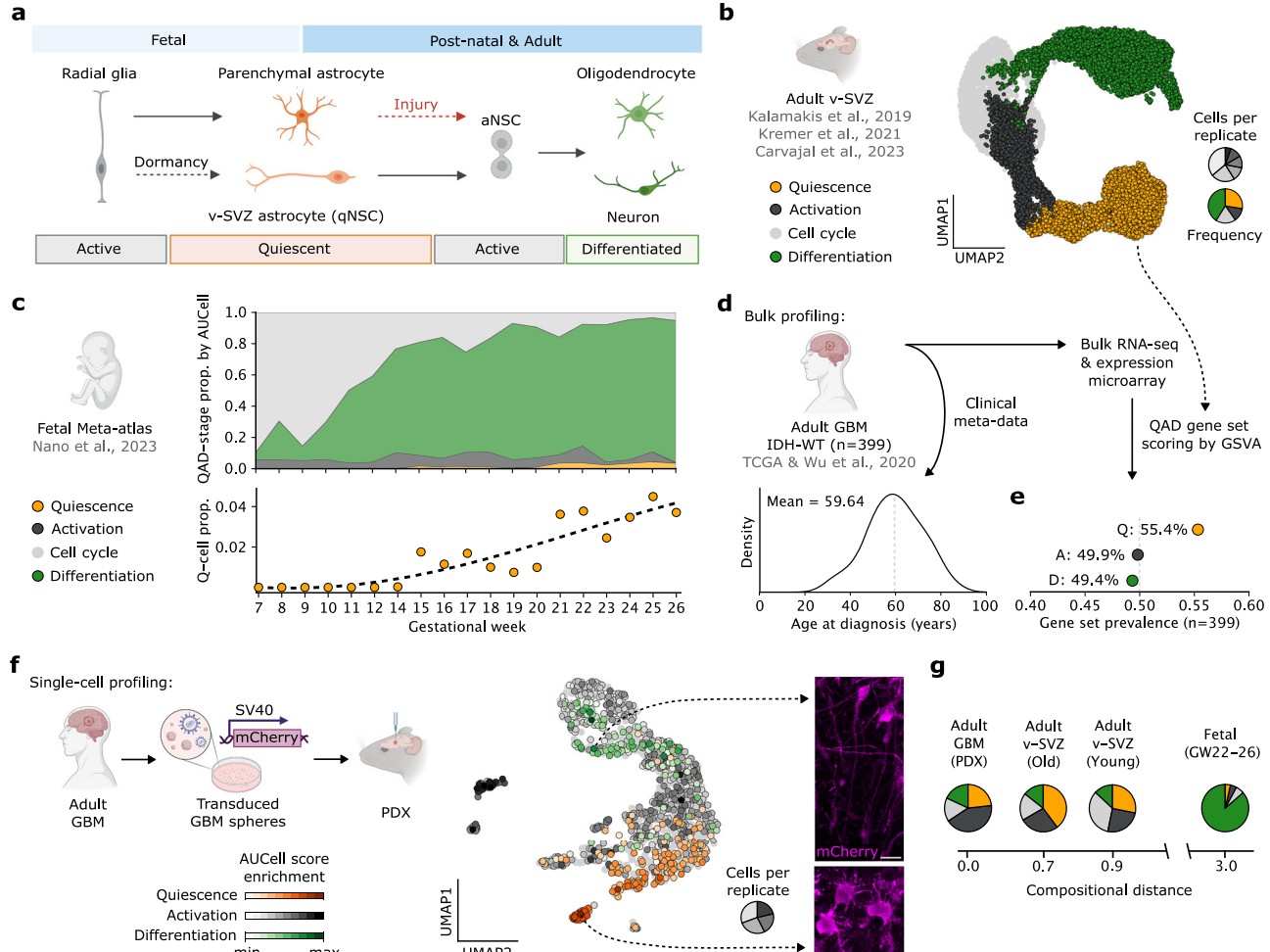

**Fig. 1 | ASA in the fetal and adult mammalian brain. a** Schematic representation of the embryonic origins of adult v-SVZ astrocytes (qNSCs) and parenchymal astrocytes along with their lineage potential in adulthood under homeostatic and injury conditions, respectively. Created in BioRender. Kaya, O. (2025) https://BioRender.com/np19rtk. **b** UMAP embedding of integrated WT mouse v-SVZ NSC lineage scRNA-seq[15,19,37] (n = 6 replicates), showing NSC differentiation trajectory through quiescence, activation, and differentiation (QAD) stages. **c** Proportion of QAD-stage cells during human cortical development[38], with Q-cell proportions shown below. Stage assignment by QAD-gene AUCell score. **d** Distribution of age at diagnosis for n = 399 primary GBMs in TCGA[40] and Wu et al.[24]. **e** Non-parametric QAD-gene set scoring of multi-modal expression measurements of GBMs from (**d**). Gene set prevalence denotes the proportion of GBMs having a positive GSVA score. **f** Left: schematic of GBM patient-derived xenograft (PDX) generation. mCherry ubiquitously labels tumor cells. Center: UMAP of single-cell GBM PDX (n = 4 replicates) showing QAD-gene set AUCell scores. Cells are colored by the maximum-scoring state, and enrichment relates to the margin between the maximum and second-highest scores. Right: Representative immunofluorescence images of PDX GBM cells with distinct neuron- and astrocyte-like morphologies (dashed arrows). Scale bars 25 μm. **g** Aitchison compositional distance between QAD-stage frequencies in GBM PDX from (**f**), young/old adult v-SVZ from (**a**), and late second trimester from (**c**). v-SVZ ventricular-subventricular zone, q/aNSC quiescent/active neural stem cell. Source data are provided as a Source Data file.

cells. The tumor ASA is finally determined by thresholding in the aligned pseudotime, as v-SVZ QAD-stages are identifiable from pseudotime alone. Applied to the T6 GBM, ptalign reported an adult-like ASA consistent with gene-set scoring, revealing a hierarchy of cells transitioning all activation states (Fig. 2a, right panel).

The approach outlined in ptalign provides several key innovations over existing trajectory alignment algorithms. Unlike traditional dynamic programming-based approaches that require two samples with known pseudotimes and produce a third, shared pseudotime[43–46], ptalign takes one or more scRNA-seq tumor samples, a reference scRNA-seq dataset, and a pseudotime-predictive gene set to yield a single, reference-based tumor pseudotime. By aligning all tumor cells along a common differentiation axis, ptalign ensures that cells at the same pseudotime exhibit comparable transcriptomic profiles across different tumors, facilitating systematic tumor-tumor and tumor-healthy comparisons. This strategy also enables the transfer of contextual information from healthy

lineage trajectories, providing important insights for the discovery of specific tumor vulnerabilities.

To validate ptalign performance, we assessed its ability to reconstruct underlying dynamics across several benchmarking scenarios. First, applying ptalign to reconstruct neurogenic lineages in our HBO dataset (Supplementary Fig. 1h) and an independent v-SVZ NSC lineage dataset showed strong correlations between ptalign-derived and dataset-derived pseudotimes (v-SVZ Pearson = 0.95; HBO Pearson = 0.85; Supplementary Fig. 3a–f). In ptalign, both the reference and aligned pseudotimes capture the same underlying process, motivating their comparison by dynamic time warping[47] (DTW). As a successful alignment matches query cells with the reference trajectory at each point in pseudotime, we can expect a characteristic, narrow, and highly correlated diagonal in the DTW matrix. We quantify the shape of this diagonal by a score-maximizing traceback, which is compared across gene-permuted ptalign runs to derive a P-value representing the statistical confidence in aligned pseudotimes (Fig. 2e and Supplementary

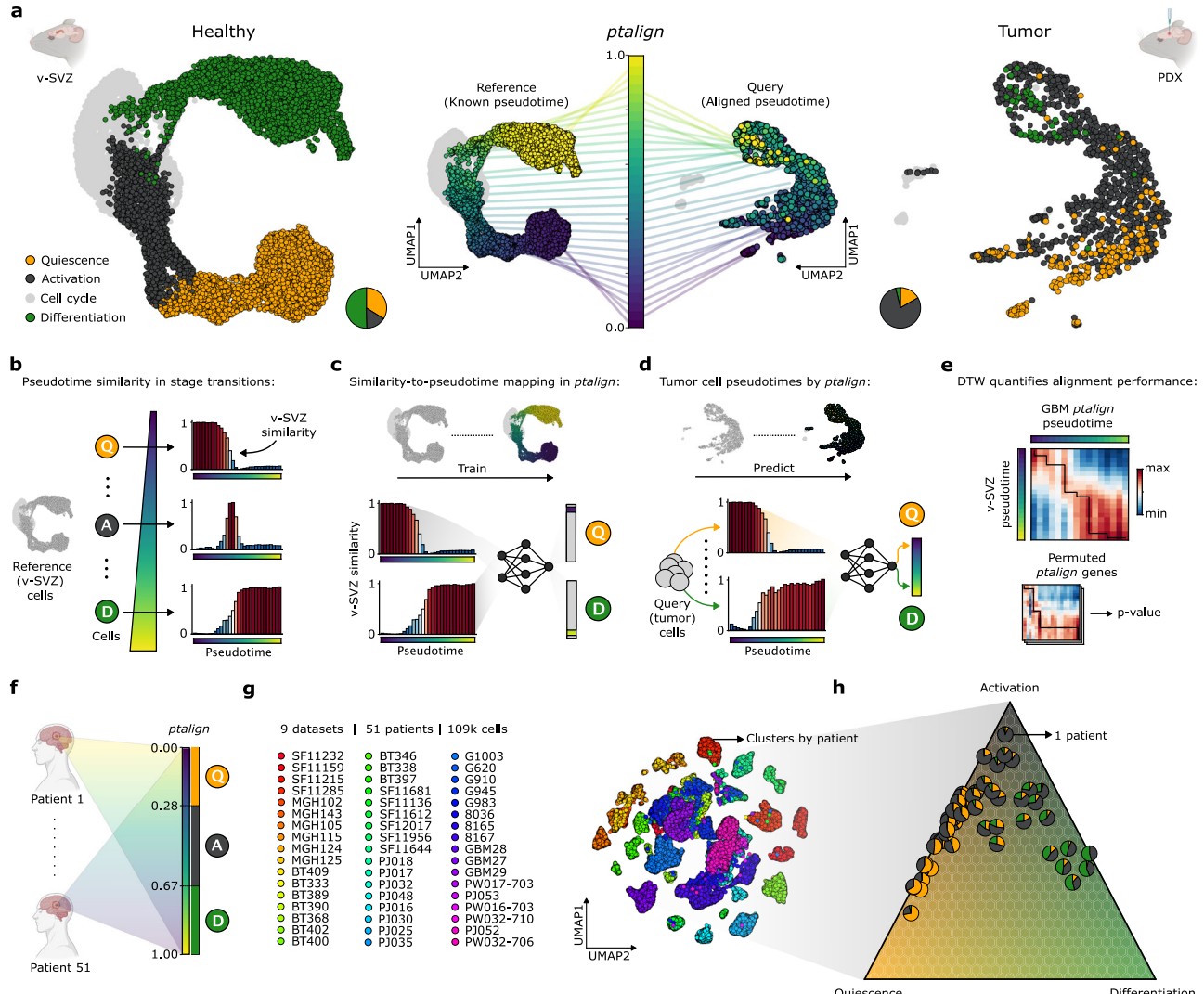

**Fig. 2 | ptalign decodes the ASA of GBMs using the adult NSC lineage as a reference. a** Left: UMAP embedding of integrated WT mouse v-SVZ NSC lineage scRNA-seq (*n* = 6 replicates), showing NSC differentiation trajectory through QAD stages. Center: v-SVZ lineage pseudotime acts as a reference into which query GBM cells are mapped by ptalign. Colored lines depict pseudotime-binned cells linked to their average position in UMAP. Right: UMAP of GBM PDX (*n* = 4 replicates) with ptalign-derived QAD-stages. Cycling cells (gray) are excluded from pseudotime analysis. Pie charts indicate QAD-stage proportions. **b** A pseudotime similarity metric derived from expression correlation along the v-SVZ lineage captures different stages and transitions by their characteristic similarity profiles. **c** In ptalign, a neural network is trained to predict v-SVZ lineage pseudotime based on similarity profiles derived from the masked reference. **d** Query tumor cell pseudotimes are assigned based on neural network predictions of v-SVZ similarity profiles. QAD-stage is derived based on the ptalign pseudotime value. **e** ptalign performance is quantified by DTW, while a permutation framework tests for alignment robustness. DTW values represent the transcriptome correlation of reference and tumor cells binned in pseudotime. **f** ptalign outlines a comparative view of tumor hierarchies, mapping patient samples within a single reference trajectory and enabling their comparison in that context. **g** scRNA-seq UMAP of *n* = 51 primary GBMs (Supplementary Data 2 and Supplementary Fig. 4), colored by patient. **h** Ternary plot of GBMs from (**g**) arranged by ptalign QAD-stage proportion, unveiling the underlying QAD-stage heterogeneity. Source data are provided as a Source Data file.

Fig. 3g; "Methods"). Applying ptalign to 55 primary GBM scRNA-seq datasets (see below) and other tissue cancers[48] revealed QAD dynamics detected in 51/55 GBMs but not in other cancers (Supplementary Fig. 3h), highlighting the sensitivity of this permutation approach. We further benchmarked ptalign against established trajectory inference methods, demonstrating comparable or better ability to reconstruct coherent trajectories (Supplementary Fig. 3i, j; "Methods"). Comparing a dataset of healthy and diseased lungs analyzed in Genes2Genes[46], ptalign not only recovered the previously reported aberrant gene dynamics, but also highlighted the involvement of TNF-alpha aberrancies in this disease[49] (Supplementary Fig. 3k–o; "Methods"). Taken together, these benchmarks demonstrate the robust and scalable framework provided by ptalign to enable ASA prediction and the comparative analysis of gene expression dynamics in healthy and disease contexts.

## ASA informs tumor growth dynamics, prognosis, and patient stratification

To interrogate inter-tumoral heterogeneity in GBM, we compiled 55 primary GBM scRNA-seq datasets[4,6,8,41,50–53] for which we determined v-SVZ-aligned pseudotimes and ASAs (Supplementary Data 2; "Methods"). 51 GBMs passed the v-SVZ ptalign permutation threshold (*p* < 0.05), and conventional UMAP clustering resolved these tumors by patient origin, obscuring underlying biological similarities (Fig. 2f, g). Instead, projecting these tumors onto a common cell trajectory using ptalign revealed a landscape of GBM ASA comprised of heterogeneous Q- and D-dominated tumors (Fig. 2h). Although clinical metadata were limited in this cohort, we found a significant association between ASA and patient age, as well as IDH-mutation status, while no clear association was detected for other features including clonal architecture[5], tumor location, or MGMT methylation status[54] (ANOVA

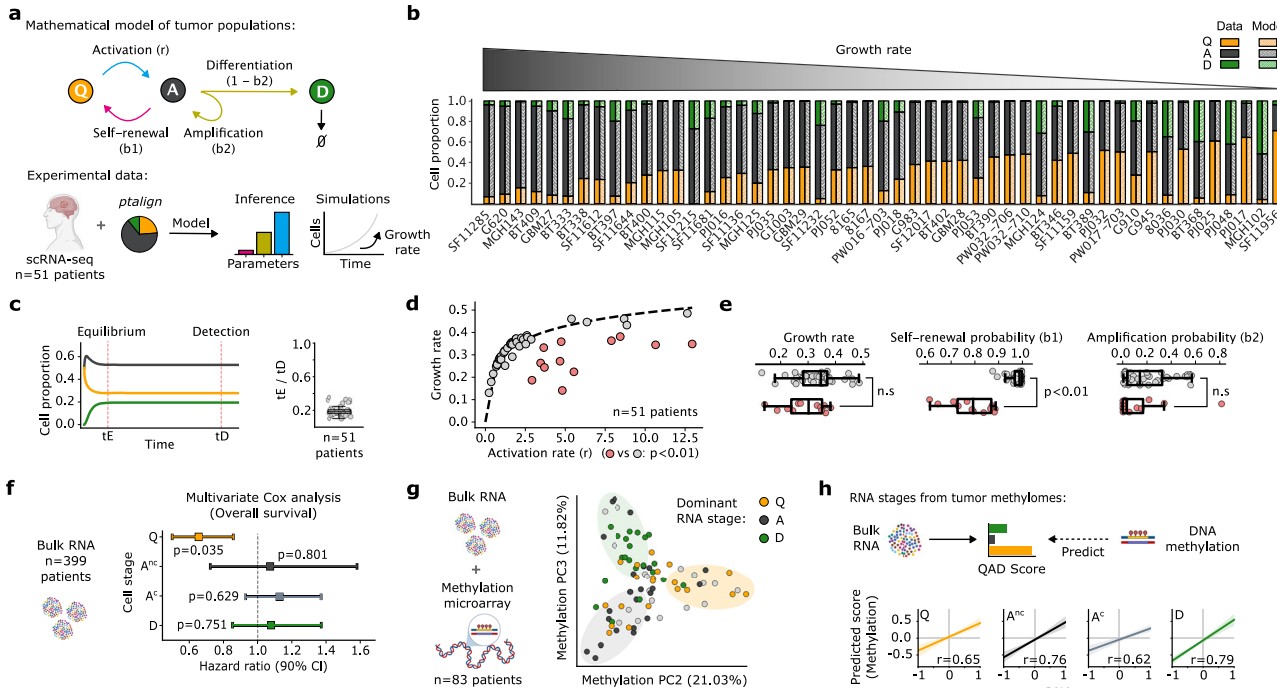

**Fig. 3 | ASA informs GBM growth dynamics, prognosis, and DNA methylation-based stratification. a, b** Application of v-SVZ population models[19,58] to GBM QAD-stage structure by ptalign, inferring descriptive parameters (**a**) and simulating growth to quantify growth rate among *n* = 51 primary scRNA-seq GBMs (**b**). **c** Left: Model simulated cell stages over time for the example tumor in (**a**). QAD populations reach steady-state equilibrium (tE) and grow until detection (tD, at 10¹¹ cells). Right: tE/tD ratios across tumors in (**b**). **d, e** Model simulated activation- and inferred growth rates for *n* = 51 primary GBMs. The dotted line shows a logarithmic fit of growth rate among GBMs (**d**). Tumors with high-residuals are shown in red, with the group-wise difference in growth rate and other model parameters shown in (**e**). Statistical significance was assessed by a two-sided *t*-test, precisely 0.11 and 0.36 for Growth rate and Amplification probability, respectively. **f** Predicted hazards with

90% confidence interval from a Cox model of *n* = 399 bulk GBMs from TCGA[40] and Wu et al.[24], with age and sex covariates of overall survival by GBM-QAD signature scores. *P*-value from multivariate Cox model with BH-correction. **g** PCA embedding of variable methylation sites for *n* = 83 GBMs from (**f**) with matched RNA and methylome measurements. Tumors are colored by their dominant RNA stage; gray circles have no clear dominant stage. Localized stage-enrichments are underscored with ellipses. **h** Linear regression over ElasticNet predictions on a (*n* = 28 GBMs) holdout set, with 90% confidence interval of RNA QAD-scores predicted from methylomes in (**g**). Models were trained on a (*n* = 55) GBM cohort. Pearson correlations (*r*) are indicated. Aⁿᶜ: active non-cycling; Aᶜ: active cycling. Box plots (**c, e**) span the 25th to 75th percentile, with the median indicated. Whiskers extend to 1.5 times the interquartile range. Source data are provided as a Source Data file.

*p* < 0.05; Supplementary Fig. 4, 5). Existing GBM metamodules from Neftel et al.[4] are assigned to their expected QAD-stages (Supplementary Fig. 6a), including a distinct Q-bias for AC-like (astrocyte-like) GBM cells and D-bias among NPC-like (neural progenitor-like) cells. Further underscoring the advantage of the ptalign mapping, we find MES-like (mesenchymal-like) GBM cells split between Q- and A-stages in line with the reported link of quiescence to hypoxic and activation to wound-response signatures in the mesenchymal population[51,55,56]. Thus, GBM ASAs decoded by ptalign offer functional insights into tumor organization, paving the way for individual patient stratification.

Next, we addressed the potential prognostic value of the decoded tumor ASAs. By integrating these data into mathematical models, we can forecast how a tumor evolves, when it might surge in growth, and how it could respond to treatment. As in vivo models of GBM hierarchy support an adult NSC-like organization rooted in a quiescent progenitor[10,20,57], we applied a system of ordinary differential equations (ODEs) adapted from established models of v-SVZ population dynamics[19,58,59] to model tumor progression (Fig. 3a and Supplementary Data 3). Fitting this model to our 51-GBM cohort, we simulated QAD population dynamics, faithfully recapitulating tumor-specific ASAs and generating quantitative estimates of tumor growth rates. This revealed a positive correlation between growth rate and the proportion of A-stage cells, which was reversed for Q-stage cells (Fig. 3b). Notably, our simulated QAD-stage proportions for all 51 GBMs rapidly reached equilibrium during early tumor development (Fig. 3c), reflecting the early establishment of growth-supporting

architectures. Though such dynamics can be expected from the class of linear ODEs applied here (see Supplementary Data 3), it is consistent with recent non-linear ODE models of the v-SVZ[59], as well as the early establishment and faithful recapitulation of cellular architectures observed in GBM organoids[41,42] (Supplementary Fig. 2). To gain insights into the processes governing the underlying tumor ASA, we next investigated the effect of our modeled parameters on tumor growth rates. The activation rate (i.e., Q–A transition, representing the exit of quiescence) was the strongest predictor of tumor growth rate (59.7% explained), while a subset of tumors compensated for lower activation rates through increased self-renewal (Fig. 3d, e). This is consistent with adult NSCs, where decreasing activating rates in older ages are compensated through increased self-renewal in an interferon-dependent manner[19]. Together, this suggests that Q–A transitions are a critical determinant of tumor growth and progression. We confirmed this observation by further decomposing 399 primary human GBMs using a 271-gene GBM-QAD signature ("Methods"; Supplementary Data 1), revealing Q-scores significantly associated with improved patient outcomes (Fig. 3f and Supplementary Fig. 6b, c). These insights highlight the role of Q-stage cells in sustaining GBM growth[25,27,57] and pinpoint the Q–A transition as a promising target for therapeutic intervention.

Collectively, our results reveal that tumor ASA has prognostic significance and informs parameters related to tumor growth, thereby guiding the identification of therapeutic targets. Consequently, developing clinical biomarkers for patient stratification based on GBM ASAs is a clear priority. Thus, we examined whether a patient's

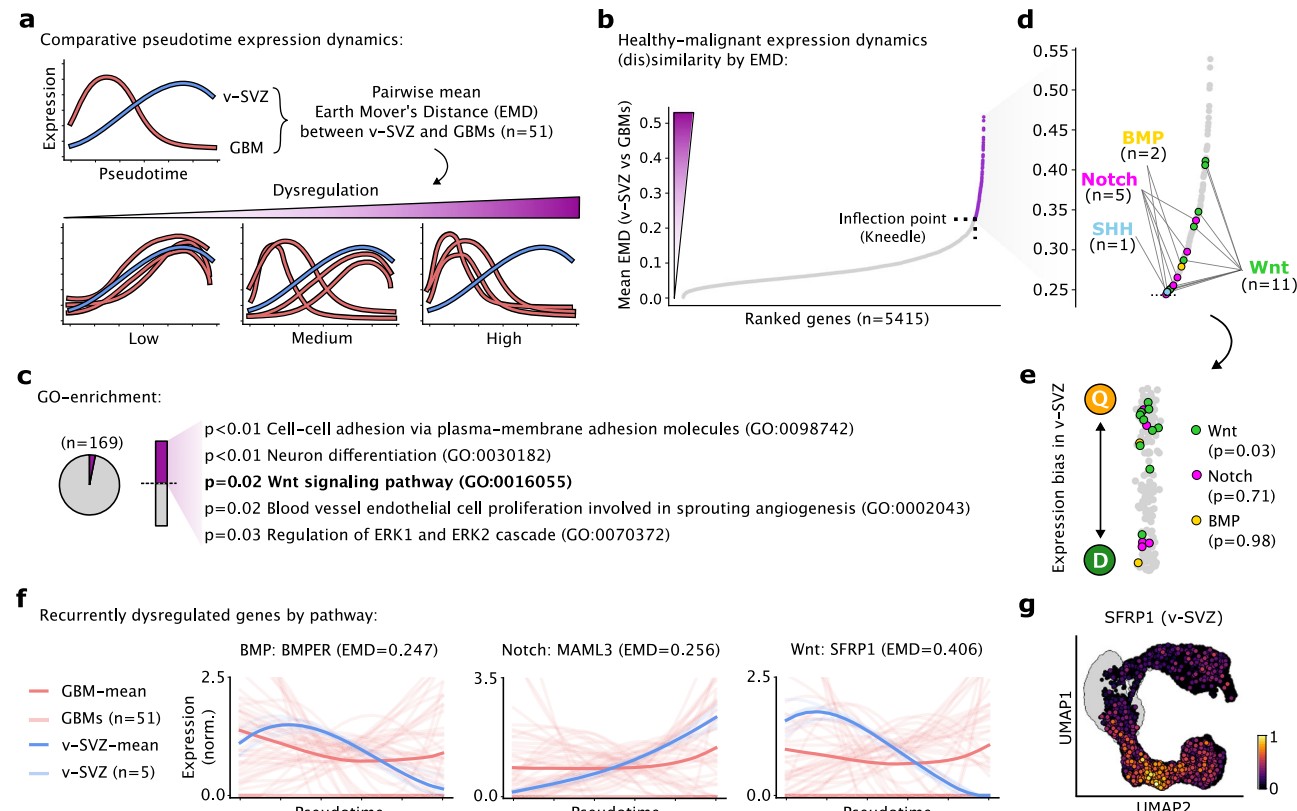

**Fig. 4 | Comparative analysis of expression dynamics reveals recurrently dysregulated pathways in GBM. a** Shared ptalign axis between GBMs and v-SVZ enables comparative assessment of healthy vs malignant expression dynamics by pairwise EMD. Mean EMD across GBMs informs gene dysregulation among the GBM cohort. **b** Genes ranked by mean EMD vs v-SVZ among $n = 51$ GBMs. Genes above the inflection point ($n = 164$, purple) are considered recurrently dysregulated. **c** GO enrichment of recurrently dysregulated genes from (**b**) with all considered genes as background. *P*-values by hypergeometric test with FDR-correction. **d** Zoomed view of recurrently dysregulated genes from (**b**) with key signaling pathway genes indicated. **e** v-SVZ expression bias for recurrently dysregulated genes from (**b**), colored by pathway genes from (**d**). *P*-values represent a one-sided permutation test for Q-bias. **f** Expression splines and EMD values for individual genes from (**d**). Individual, and 95% confidence intervals, of mean GBM expression dynamics are colored red, while v-SVZ dynamics are blue. **g** Log-normalized SFRP1 expression in the v-SVZ UMAP. Cycling cells are colored gray. Source data are provided as a Source Data file.

methylation profile correlates with their ASA, since QAD-stages of the v-SVZ NSC lineage display distinct methylomes[30] and such profiling is already employed in the WHO-classification of brain tumors[60]. Analysis of 83 IDH-wildtype tumors with matched RNA and methylation profiles revealed that a PCA of methylation data decomposed tumors according to their dominant RNA-defined QAD-stage (Fig. 3g and Supplementary Data 4). We found that inter-tumoral RNA- and methylation-distances were correlated, motivating us to train an ElasticNet model to predict RNA scores from tumor methylomes (Supplementary Fig. 6d; "Methods"). These predictions achieved a mean Pearson correlation of 0.70 on holdout samples (Fig. 3h), demonstrating that GBM ASA can be inferred from bulk methylomes. Taken together, our modeling and biomarker analyses reveal the central role of Q-stage cells and the Q–A transition in shaping patient outcomes, motivating methylation-based biomarkers for patient stratification by ASA.

**Target discovery by comparative analysis of dysregulated expression dynamics**

By projecting tumor cells onto a shared reference trajectory using ptalign, we enable direct comparisons of healthy vs malignant expression dynamics. Current approaches for detecting differential gene expression along pseudotime are chiefly sensitive to differences in expression levels, which are frequently altered in tumors and can overshadow changes in the underlying expression dynamics. To overcome this limitation, we normalize the area under pseudotime expression curves and apply the Earth Mover's Distance[61] (EMD) to quantify differences in curve shape that represent differential regulation along the underlying trajectory. This way, genes with similar dynamics obtain lower EMD scores while those with divergent expression patterns obtain high EMD scores (Fig. 4a). We use this approach to identify putative biomarkers, characterized by consistent expression dynamics and changes in overall expression, as well as therapeutic targets, which exhibit consistently altered expression dynamics.

To explore actionable biomarkers for patient stratification, we computed pairwise EMDs for two groups of Q- and D-dominated GBMs (Supplementary Fig. 6e, f and Supplementary Data 5). This analysis highlighted SOX11 and STMN1 as potential biomarkers with consistent expression dynamics independent of ASA, which also exhibited group-specific expression levels and methylation patterns (Supplementary Fig. 6g, h). In contrast, genes including KLF4 and ZFP36 exhibited differential expression dynamics by group that were not consistently echoed by their group-specific expression levels (Supplementary Fig. 6i, j), thus undermining their prognostic value and instead being relevant predictors of the tumor biology.

We next compared EMD values between GBMs and the v-SVZ reference to identify genes with consistently aberrant dynamics, which could serve as therapeutic targets. This required pre-selection of genes universally expressed in both healthy and malignant contexts, revealing 169 genes with consistently dysregulated expression dynamics in GBM (Fig. 4b and Supplementary Data 5; "Methods"). These genes are enriched in processes commonly disrupted in GBM, including cell-cell adhesion, neuronal differentiation, angiogenesis, and Wnt signaling[22]

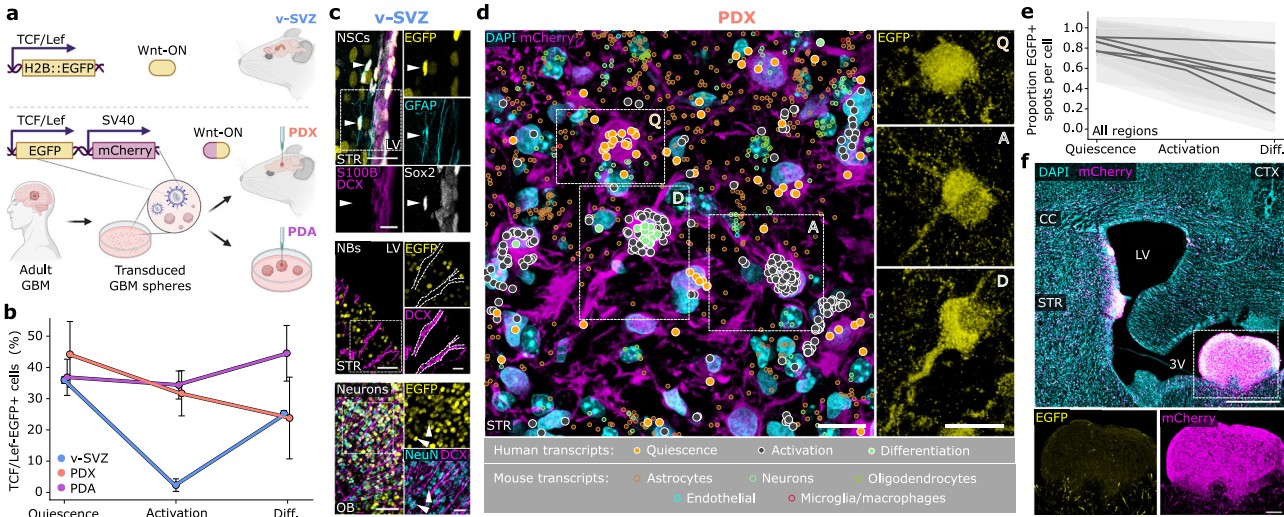

**Fig. 5 | Dysregulation of Wnt activity at the Q–A transition in GBM. a** TCF/Lef-EGFP construct reporting canonical Wnt signaling activity in v-SVZ NSCs from a transgenic mouse line (top) and by lentiviral vector (bottom) in PDX and PDA tumors. mCherry ubiquitously labels tumor cells. Created in BioRender. Kaya, O. (2025) https://BioRender.com/np19rtk. **b** TCF/Lef-EGFP activity quantified in QAD-stage v-SVZ NSCs ($n = 1564$ cells, $n = 5$ replicates) and T6 PDX ($n = 1176$ cells, $n = 4$ replicates), and PDA ($n = 12,378$ cells, $n = 4$ replicates) cells. Reporter activity was quantified by FACS and QAD-stage by scRNA-seq. Data presented as mean, error bars are standard deviation in the normalized Wnt-active cell proportion. **c** Representative immunofluorescence images of TCF/Lef-H2B::EGFP activity for v-SVZ lineage populations. Top: EGFP+ NSCs (arrow heads) lining LV identified by presence of GFAP, SOX2, and absence of S100B and DCX (scale bars 20 μm and 10 μm). Middle: EGFP- NBs (Dotted lines; scale bars 50 μm and 20 μm) in v-SVZ marked by the presence of DCX. Bottom: EGFP+ neurons (arrow heads) marked by

the presence of NeuN in OB (scale bars 50 μm and 20 μm). **d** Representative spatial transcriptomics view of QAD-stage cells in a GBM PDX. Left: striatal section of clustered cells from QAD-stages overlaid with DAPI and mCherry immuno-fluorescence. Tumor transcripts are colored by QAD-stage with a white outline; mouse transcripts are colored by cell type. Right: QAD-stage cell EGFP fluorescence. Scale bars, 10 μm. **e** Mean fraction of EGFP+ spots for QAD-stage cells in $n = 6$ spatial transcriptomics ROIs. **f** Immunofluorescence image of GBM PDX tumor cells (mCherry) in a large ventricular outgrowth of the third ventricle. Scale bars 500 μm and 100 μm. Insets highlight tumor cells devoid of Wnt activity, which is regained upon entry to the brain parenchyma. NB neuroblast, STR striatum, LV lateral ventricle, OB olfactory bulb, CTX cortex, CC corpus callosum, 3V third ventricle. Each experiment was repeated independently at least three times. Source data are provided as a Source Data file.

---

(Fig. 4c). Indeed, Wnt pathway genes outnumbered those from BMP, Notch, or SHH signaling pathways among our GBM dysregulated genes (Fig. 4d). These Wnt genes in turn showed enriched Q-stage expression in the v-SVZ lineage (Fig. 4e). Among these, SFRP1 was one of the most consistently dysregulated genes, notably peaking at the Q–A transition in v-SVZ NSCs (Fig. 4f, g). Thus, SFRP1 and other genes may provoke aberrant Wnt dynamics at the Q–A transition, contributing to GBM progression.

To further investigate the role of dysregulated Wnt signaling in GBMs, we measured canonical Wnt (cWnt) activity levels across QAD-stages in both healthy and tumor lineages using genetic reporters (TCF/Lef-EGFP; "Methods") (Fig. 5a). Analysis of the v-SVZ NSC lineage in this cWnt reporter line revealed tight regulation of Wnt activity across QAD-stages: cWnt was highest at the Q-stage, decreased in the A-stage, and resurged in the D-stage (Fig. 5b, c and Supplementary Fig. 7a). In contrast, GBM cells exhibited sustained cWnt-reporter activity across all QAD stages (Fig. 5b). This was confirmed by spatial transcriptomics of a cWnt-reporter PDX, revealing consistent cWnt activity across brain regions (Fig. 5d, e and Supplementary Fig. 7b–d; Supplementary Data 6) with one notable exception: ventricular outgrowths (Fig. 5f). Notably, these outgrowths, as well as in vitro cultured GBM spheroids lack cWnt activity (Supplementary Fig. 7e–h), possibly due to their lack of contact with brain parenchyma[62]. Intriguingly, ventricular out-growths were characterized by A- and D-stage cells organized along distinct gradients with Q cells lying at the ventricular wall (Supplementary Fig. 7i). Overall, our comparative approach reveals that GBMs lose the tight regulation of cWnt signaling that accompanies the exit from quiescence in healthy NSCs, prompting further investigation of dysregulated factors involved in this transition.

## SFRP1-induced quiescence stalls tumor growth

Using ptalign, we identified SFRP1, a secreted Wnt antagonist[32], as one of the most consistently dysregulated genes in GBM (Fig. 4f). Thus, to disrupt tumor ASA by modulating cWnt signaling at the Q–A transition, we cloned SFRP1 downstream of the TCF/Lef-promoter in T6 GBM (Fig. 6a and Supplementary Fig. 8a, b; "Methods"). Strikingly, SFRP1-overexpression (OE) in tumor cells significantly improved overall survival in mouse PDXs, while scRNA-seq revealed that GBM cells were stalled in a quiescent astrocyte-like stage (Fig. 6b–d). Applying our GBM population models revealed that the stark reduction in tumor growth rate resulted from a decrease in activation rate (Fig. 6e), pinpointing the action of SFRP1 to the Q–A transition. These results were mirrored by the OE of NOTUM, another recurrently dysregulated cWnt antagonist[63] expressed at the Q–A transition (Supplementary Fig. 8c–e). NOTUM-OE and SFRP1-OE tumors exhibited similar transcriptional profiles (Supplementary Fig. 8f; Pearson = 0.76), suggesting these converged on a set of overlapping downstream effectors.

The astrocytic shift of SFRP1-OE transcriptomes was accompanied by a dramatic morphological transformation from a neuron-like appearance in GBM cells homing to the cortex to the star-shaped form typical of astrocytes (Fig. 6f). We further employed spatial transcriptomics to explore the distribution of QAD-stage cells in SFRP1-OE tumors. This revealed that tumor cells across all brain regions were confined to the Q-stage (Fig. 6g), while neigh-borhood enrichment highlighted their preference for neuronal proximity (Supplementary Fig. 8g). The only exception was found in SFRP1-OE ventricular outgrowths, which organized along A-D gradients (Supplementary Fig. 8h) as previously, as the lack of cWnt activity impairs the action of SFRP1, thereby additionally

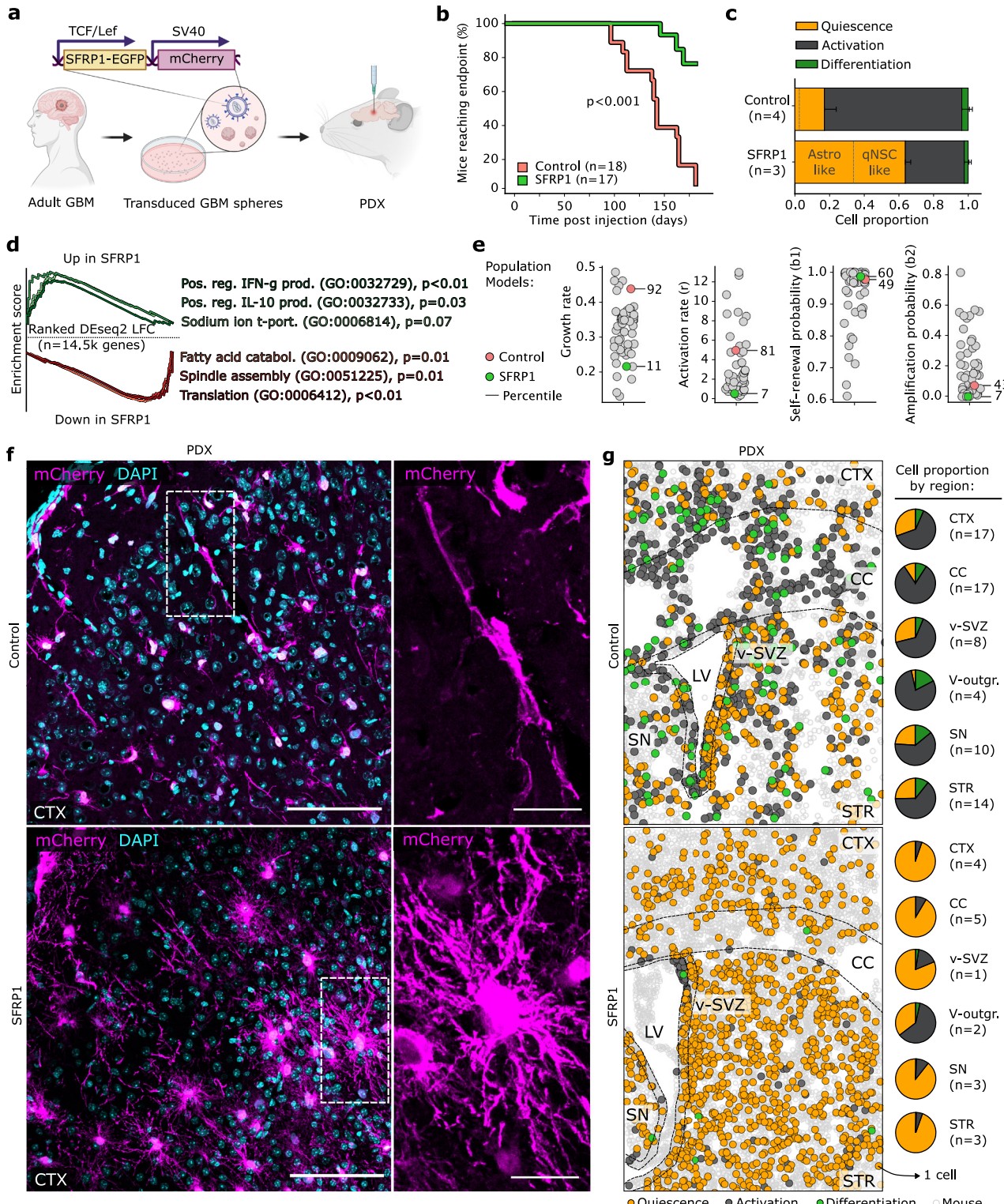

**Fig. 6 | Targeted disruption of ASA by SFRP1 renders the tumor quiescent.**
**a** SFRP1-OE lentiviral construct used to generate GBM PDXs. mCherry ubiquitously labels tumor cells. Created in BioRender. Kaya, O. (2025) https://BioRender.com/np19rtk. **b** Kaplan–Meier curve of mice reaching endpoint post injection among three batches of $n = 6$ for control and SFRP1-OE mice each. *P*-value from log-rank test, precisely $1.3 \times 10^{-4}$. **c** Proportion of QAD-stage cells identified by ptalign in SFRP1-OE ($n = 3$ replicates) and control ($n = 4$ replicates) scRNA-seq. Bars present mean, error bars standard deviation. **d** Selected GSEA enrichments from genes ranked by DEseq2 log fold-change between pseudobulked SFRP1-OE and control from (**c**). *P*-values from GSEA enrichment are FDR-adjusted. **e** Parameter estimates from population models of GBM dynamics (Fig. 3), including activation- and

inferred growth rates, as well as self-renewal and amplification probabilities for T6 control and SFRP1-OE GBM PDXs from (**c**) among $n = 51$ primary GBMs. The rank percentile for each parameter is indicated. **f** Representative immunofluorescence images of GBM cells in a control and SFRP1-OE (**e**) PDX brain. Scale bars 100 μm, in insets 25 μm. **g** Entire spatial transcriptomics ROI depicting similar regions in SFRP1-OE (**g**) and control PDX brains. Transcripts were associated with segmented nuclei to assign species and QAD-stage. Pie charts indicate the sum of QAD-stage cells by brain region across ROIs. ROI region of interest, CTX cortex, CC corpus callosum, LV lateral ventricle, V-outgr. ventricular outgrowth, SN septal nuclei, STR striatum. Source data are provided as a Source Data file.

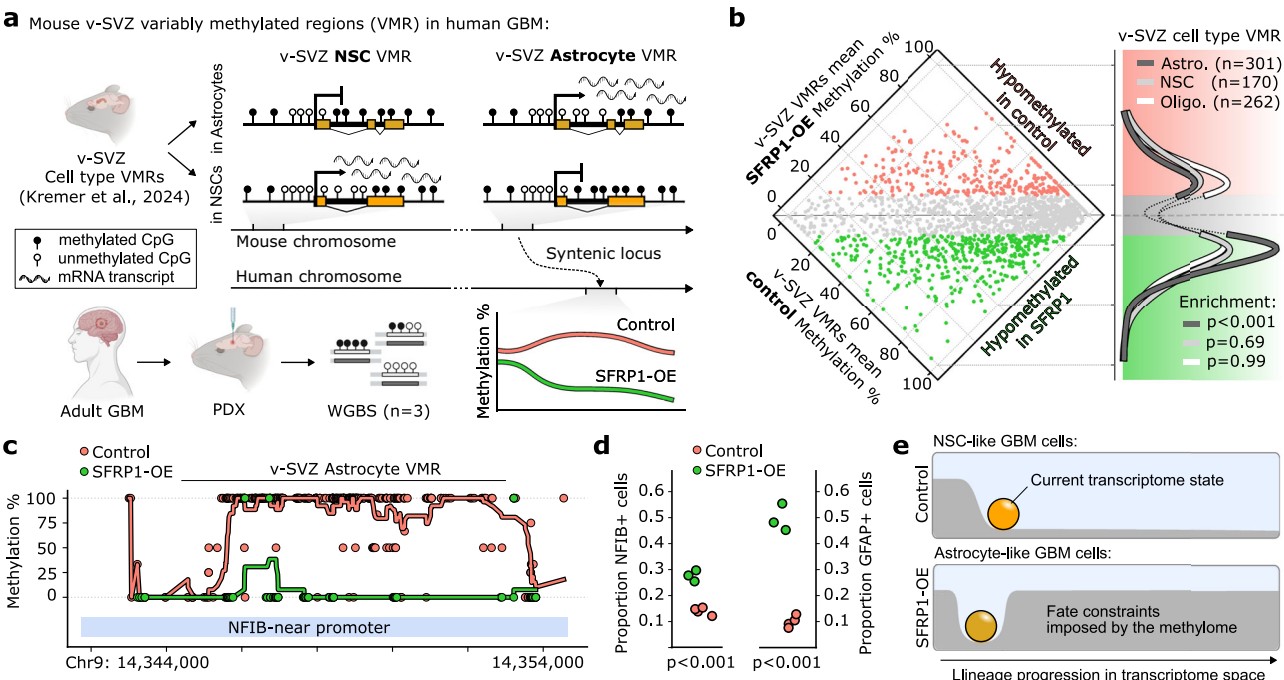

**Fig. 7 | Emergence of a conserved astrocyte-like methylome in GBM cells by SFRP1. a** Cross-species interrogation of murine v-SVZ cell type VMRs from Kremer et al.[30]. Example NSC and astrocyte VMR methylation profiles are depicted, with methylation at the corresponding human locus quantified in SFRP1-overexpressing (OE) and control GBM WGBS. **b** Left: Mean methylation of v-SVZ cell type-specific VMRs from ref. [30] in SFRP1-OE and control WGBS ($n = 3$ technical replicates each). Differentially methylated regions are highlighted by genotype. Right: v-SVZ cell type VMRs in a Gaussian KDE over the vertical axis of the scatterplot. *P*-value by one-sided hypergeometric test, for Astro. precisely $5.8 \times 10^{-5}$.

**c** Selected v-SVZ astrocyte VMR overlapping an NFIB-promoter in SFRP1-OE and control. Points depict mean CpG methylation among replicates, lines comprise a 10-CpG moving average. **d** Proportion of NFIB+ (left) and GFAP+ (right) Q-cells among control and SFRP1-OE samples. *P*-value by two-sided *t*-test, precisely $1.4 \times 10^{-4}$ and $3.4 \times 10^{-5}$ for NFIB, GFAP, respectively. **e** Schematic representation of the lineage potential in NSC-like GBM cells in control (above) vs lineage restrictions imposed by the remodeled methylome of the expanded astrocyte-like GBM cells upon SFRP1-OE (bottom). Adapted from ref. [30], Springer Nature, Inc. VMR: variably methylated region. Source data are provided as a Source Data file.

underscoring the limited range of secreted Wnt factors[64]. Indeed, while tumor cells exhibited increased quiescence in an SFRP1-OE PDA model, healthy cell transcriptomes remained unperturbed (Supplementary Fig. 8i–k).

In a recent study, we demonstrated that two functionally distinct cells, dormant v-SVZ NSCs and astrocytes, exhibit similar transcriptomes but differ in their DNA-methylome[30]. Notably, an ischemic injury to the brain remodels the astrocyte-methylome into an NSC-like one, and thereby induces stemness. As SFRP1-OE in GBM cells induced an astrocyte morphology and functional quiescence, we asked to what extent this methylome remodeling is conserved across species and disease contexts. We analyzed our previously reported v-SVZ variably methylated regions[30] (VMRs) in three technical replicates of SFRP1-OE and control PDX tumors (Fig. 7a). Analysis of v-SVZ VMRs at their syntenic human coordinates ("Methods"; Supplementary Data 4) revealed significant enrichment of astrocyte-specific VMRs hypomethylation in SFRP1-OE tumor cells (Fig. 7b), confirming that SFRP1-OE rewires tumor cell methylomes along with their transcriptomes and morphology. Notably, one of the most divergently methylated VMRs overlapped an NFIB-promoter (Fig. 7c): its product, NFIB, is the primary transcriptional activator of the astrocyte marker GFAP[65] and is also used to reprogram iPSCs into astrocytes[66]. Consistently, we identified a significant increase in NFIB and GFAP transcript-expressing cells in SFRP1-OE tumors (Fig. 7d). As methylation generally acts to restrict cellular plasticity[67], the significant degree of conserved remodeling in SFRP1-OE tumor astrocytes might render them refractory to subsequent reactivation (Fig. 7e). Collectively, our findings demonstrate the unique advantages of decoding ASA in solid tumors to enable target identification and effectively disrupt tumor growth.

## Discussion

Tumors retain characteristics of their tissue-of-origin[68], forming hierarchies that mirror developmental lineage maps[69]. Here, we leveraged the similarities between GBMs and adult NSCs to decode ASAs using ptalign, a computational method that aligns tumor cells within a reference pseudotime axis. We show that this architecture reflects an early equilibrium state from which key parameters governing tumor growth can be inferred. Together, ASAs serve as prognostic markers, support patient stratification, and inform personalized therapeutic interventions.

Central to our findings is the ability for ptalign to identify cells transitioning between activation states, representing critical bottlenecks for therapeutic intervention. This is achieved through a reference-based approach, which maps tumor cells within a healthy lineage trajectory, enabling the transfer of contextual knowledge and simultaneous comparison of tumors with both other tumors and healthy populations. Although ptalign currently supports only non-branching trajectories, emerging methods tackling branched alignments[70] and DTW[71] could extend its applicability.

By assessing activation states in bulk and single-cell GBMs, alongside transcriptomic and DNA methylation contexts, we gained insights into the ASAs that support tumor growth. Quiescence was a ubiquitous feature of GBM ASAs, with most tumors more closely aligned to adult than fetal NSC ASAs. While both fetal and adult NSC populations exhibit a similar diversity of cellular fates, quiescence is largely absent during fetal development[11]. Using mathematical models of GBM population dynamics, we identified the Q–A transition as central to tumor growth. This aligns with in vivo studies of GBM that report a tumor hierarchy rooted in a quiescent progenitor[10,20,57]. However, this organization is challenged in vitro[72,73], possibly due to

the growth-factor-enriched conditions that lead to a fetal ASA, limiting their ability to capture quiescent cell dynamics. Despite these differences, the activation of therapy-resistant quiescent cells to repopulate the tumor[25,26] underscores the clinical relevance of this state, as well as the need to devise strategies targeting this Q–A transition. Importantly, linear ODE models of cell population dynamics capture average behavior at the population level, smoothing out individual cell variability. While single-cell trajectories may be non-linear, these models abstract away such complexity to describe aggregate trends.

NSCs exhibit an increasing quiescence-bias with age that correlates with increased expression of SFRP5[15,16]. Mathematical modeling identified a reduced rate of activation that increases upon SFRP5 neutralization[15]. In contrast, our mathematical modeling of GBM shows an elevated activation rate, which can be suppressed by overexpressing SFRP1. These models also suggest that while GBM growth in most patients is driven by changes in activation rate, a subset of tumors with very low activation rates maintain growth through self-renewal, a mechanism similarly observed in the aging healthy brain upon deletion of interferon receptors[19]. Further studies are thus needed to explore signaling pathways regulating compensation through self-renewal. SFRP1 overexpression may be particularly beneficial for recurrent GBMs, which are more aggressive and exhibit a proneural shift[74].

Through ptalign-driven comparisons, we identified pervasive dysregulation of cWnt pathway genes in GBM. The dysregulated Wnt antagonists SFRP1 and NOTUM were found at the Q–A transition, and independent overexpression of either induced a pronounced astrocytic shift in tumor cell morphology, transcriptome, and methylome. The resulting methylome changes mirrored remodeling observed in murine brain regeneration, where parenchymal astrocytes undergo a switch to an NSC-like state following ischemic injury[30]. This convergence of methylome remodeling between murine regeneration and human GBM suggests that conserved epigenetic mechanisms may also underlie human brain regeneration. These findings underscore the reciprocal developmental parallels between healthy and malignant tissues, advancing a comparative framework that informs both tumor biology as well as fundamental processes regulating healthy cells. Because methylation constrains cellular plasticity by restricting the genome to the specific functions of terminally differentiated cells[67], future work will determine whether SFRP1-mediated methylome remodeling enduringly locks GBM cells in a quiescent state, offering a potential therapeutic avenue. This contrasts with the transient increase in quiescence observed following chemotherapy- or radiotherapy-induced depletion of the active tumor population[25,26], which ultimately leads to aggressive tumor recurrence.

The quiescent phenotype observed in SFRP1-OE tumor cells is consistent with its role in healthy NSCs, where knockdown of SFRP1 increased activation at the cost of depletion of the quiescent NSC pool[32]. Crucially, our strategy in SFRP1-OE targets a transition over a stage, enabling an effective disruption of tumor dynamics. SFRP1 effectively synchronized the entire tumor population in a Q-stage, notably without perturbing healthy cells, paving the way for combination therapies that complement existing treatments to improve patient outcomes.

Few molecular biomarkers reliably predict GBM therapy outcomes[54], though methylation has proved a robust and cost-effective means for tumor classification[60,75]. We identified cellular activation states encoded in tumor methylomes, which we used to predict ASA signatures from bulk samples. However, while SFRP1-OE methylome remodeling was found in WGBS, it was not detectable on commercial methylation microarrays owing to their limited CpG coverage, mainly featuring promoter regions that do not inform on stem cell trajectories[30]. We additionally demonstrated that biomarker efficacy can be compromised by stage-specific expression dynamics, and

identified SOX11 and STMN1 as potential markers with stable expression dynamics that provide transcriptome and methylation-based means to identify, e.g., those patients who are likely to benefit from SFRP1 therapy. Of note, the EMD-based approach that identified these markers in Q- vs D-dominated GBMs extends to any two patient groups, motivating the development of tailored interventions.

Our GBM PDXs revealed that ventricular outgrowths consistently lacked Wnt activity and harbored large numbers of active, cycling cells, aligning with the aggressive clinical course observed in patients with ventricular tumors[76]. Wnt activity in tumor cells may depend on interactions with the surrounding brain parenchyma[62], potentially through synaptic neuron-tumor networks[77]. Likewise, the cerebrospinal fluid, which contacts the ventricles and contains SHH ligands known to antagonize Wnt signaling[78,79], may drive the increased activation observed in ventricular outgrowths[80]. Notably, the absence of quiescent cells in these outgrowths mirrors the behavior of metastatic cells at the vessel lumen, where cWnt activity is only gained upon extravasation and triggers a proliferative-to-dormant switch[81]. These findings underscore the critical influence the microenvironment has on shaping tumor phenotypes[22].

Overall, our study provides a systematic framework for patient stratification and therapeutic target identification, emphasizing the importance of targeting dynamic tumor processes to disrupt tumor progression.

## Methods

### Experimental models

**Culture of human 293T cells.** The 293T cell line was used for the production and titration of the lentiviral particles. The 293T cell line is a fast-growing, highly transfectable derivative of human embryonic kidney 293 cells. Due to the presence of SV40 large T antigen, they allow high levels of proteins to be expressed; therefore are ideal for generating high-titer lentiviral particles.

The 293T cells were maintained in Dulbecco's Modified Eagle's Medium (DMEM) supplemented with 10% FCS, 1% penicillin/streptomycin (10,000 units/ml), and 1% L-glutamine (200 mM) at 37 °C and 5% $CO_2$.

### Culture and maintenance of human iPSCs

The human iPSCs were acquired from the Allen Institute for Cell Science (AICS-0036-006, https://www.allencell.org/cell-catalog.html). The AICS-0036-006 line was labeled with EGFP via the CRISPR/Cas9 system, where the mEGFP transgene was knocked into chromosome 19q13-qter. Their parental cell line (https://www.coriell.org, GM25256) was acquired from Coriell (UCSFi001-A, https://labs.gladstone.org/conklin/) and was derived from a healthy donor using an episomal reprogramming method[82].

iPSCs were feeder-independent, and Matrigel (Corning, 354277) was employed to coat the plates or dishes. iPSCs were maintained on mTeSR1 medium (Stem Cell Technologies, 05850 or 100-0276) at 37 °C and 5% $CO_2$. ReLeSR™ (Stemcell Technologies, 05872) was used for passaging cells, with media changes every other day.

### Patient tumors and tumor sphere culture

The primary tumor sample was received from the University Hospital Ulm upon obtaining informed consent prior to surgery. Experiments involving resected patient tumors were carried out in accordance with the Declaration of Helsinki and were approved by the ethics commission of the Medical Faculty Heidelberg (S-224/2021). All tumor specimens were examined by a neuropathologist to ensure that the tumors met GBM criteria defined by the World Health Organization. Upon arrival, fresh tissues were immediately dissociated using the Brain Tumor Dissociation Kit (P; Miltenyi) and expanded in culture. This includes the male T6 patient. Tumor spheres were maintained in serum-free Neurobasal A medium

supplemented with B27, heparin (2 µg/ml), and the stem mitogens EGF (20 ng/ml) and bFGF (20 ng/ml) at 5% $CO_2$ and 37 °C. For passaging, tumor spheres were enzymatically dissociated into single cells using Accutase when sphere size reached approximately 100 µm in diameter, approximately once a week.

## Mouse strains
Male TCF/Lef:H2B/GFP mice[83] were bred at the Center for Preclinical Research at the German Cancer Research Center and were used for analyzing TCF/Lef activity in mouse NSC studies. Male Fox Chase SCID Beige mice (CB17.Cg-Prkdc$^{scid}$Lyst$^{bg-J}$/Crl) were purchased from Charles River and were used to generate human-mouse xenograft tumors. Experimental mice had *ad libitum* access to food and water and were housed in specific pathogen-free, light (12 h day/night cycle), temperature (21 °C), and humidity (50–60% relative humidity) controlled conditions. All procedures were approved and conform to the regulatory guidelines of the official committee (Regierungspräsidium Karlsruhe, Germany; G19/21).

## Lentiviral transduction
Human GBM cells were transduced with the 7TGC (Addgene plasmid #24304), 7TGC-SFRP1, or 7TGC-Notum lentiviral vectors at a multiplicity of infection (MOI) of 5. Expression of the transduction marker (mCherry) was confirmed by FACS analysis. Second-generation, replication-deficient lentiviral particles were produced, purified, and titrated as described previously in ref. 84.

## Immunocytochemistry
For 2D culture of human GBM cells, $3 \times 10^5$ cells in 0.3 ml of Neurobasal A medium were seeded into each well of µ-Slide 8-well chamber (Ibidi). Prior to seeding, the wells were coated with 10 µg/ml poly-D-Lysine overnight at room temperature, washed three times with DPBS, and subsequently coated with 10–20 µg/ml Laminin for 2 h at 37 °C. The cells were treated with a recombinant Wnt3a (200 ng/ml dissolved in 0.1% bovine serum albumin (BSA) in distilled water). 0.1% BSA was used in carrier-only conditions as baseline controls. Following 24 h post-treatment, the cells were washed three times with DBPS and fixed with 2% PFA for 20 min at room temperature. Next, the fixed cells were incubated in permeabilization/blocking solution (0.25% Triton-X 100 in PBS supplemented with 3% horse serum and 0.3% BSA) for 1 h at room temperature to block unspecific binding sites. Subsequently, cells were stained with anti-GFP antibody (Supplementary Data 7) in blocking buffer overnight at 4 °C. After washing three times 5 min with 0.25% Triton-X 100 in PBS, secondary antibody staining was carried out using appropriate Alexa-fluorophore antibodies (Supplementary Data 7) diluted in blocking buffer for 1 h at room temperature. DAPI was added to the antibody cocktail for nuclear staining. Furthermore, cells were washed three times 15 min with 0.25% Triton-X 100 in PBS. After mounting with Fluoromount-G, the slide was allowed to dry in the dark at room temperature for 30 min. Samples were stored in the dark at 4 °C until imaging.

For 3D culture of human GBM cells, individual tumor spheres of approximately 100 µm in diameter were seeded in 0.1 ml of Collagen matrix containing 0.27 mg of Collagen type I (Corning; rat tail) supplemented with 10% minimum essential medium (MEM; 10×) and 20% NaOH (0.1 M) for pH adjustment in a well of 10-well cell view cell culture slide (Greiner Bio-one). Collagen matrix was allowed to polymerize for 1 h at 37 °C, and 0.3 ml of Neurobasal A medium with recombinant Wnt3a (200 ng/ml in 0.1% BSA in distilled water) or carrier-only (0.1% BSA in distilled water) was added for 24 h before live-imaging.

Confocal images were acquired on a Leica TCS SP5 or SP8 confocal microscope at the light microscopy facility of the German Cancer Research Center. ImageJ was used for processing, and images were only adjusted for brightness and contrast.

## Molecular cloning of 7TGC-SFRP1 and 7TGC-Notum constructs
Human SFRP1 (Origene plasmid #RC207328) or Notum (Gateway ORF clone ID #164485821) was cloned in frame under 7xTCF promoter (7TGC; Addgene plasmid #24304) upstream of EGFP sequence using In-Fusion HD cloning kit (Takara) according to the manufacturer's instructions (Supplementary Data 7). A furine cleavage site (RKRR), as well as a self-cleaving peptide linker (P2A), was incorporated between SFRP1/Notum and EGFP to allow seamless protein translation, avoiding fusions (Supplementary Fig. 8c, i). 7TGC plasmid was a gift from Roel Nusse. P2A-RKRR plasmid was a gift from Edward Green. Notum plasmid was obtained through the vector and clone repository[85] of the genomics and proteomics facility of the German Cancer Research Center. Constructs cloned as part of this study are available from the investigators upon request.

## Enzyme-linked immunosorbent assay (ELISA) for SFRP1
T6 Wnt-reporter or SFRP1-OE GBM cells ($5 \times 10^5$ per well) were seeded in a 6-well plate in Neurobasal A medium. The cells were treated with a recombinant Wnt3a (dissolved in 0.1% BSA in distilled water at a final concentration of 200 ng/ml). The 0.1% BSA was used in carrier-only conditions as baseline controls. Following 24 h post-treatment, the supernatant was collected and the ELISA protocol was followed according to the manufacturer's instructions (Abcam; ab277082). The optical density (OD) of the samples was measured at 450 nm with the plate reader (Agilent Biotek). The respective concentration of each sample was calculated based on the standard curve generated by known protein concentrations and measured OD at 450 nm following background subtraction according to the manufacturer's instructions (Abcam; ab277082).

## Human brain organoid generation
The procedure was adapted from a previously published method[39,86,87]. Briefly, on day 0, hiPSCs were first dissociated into single cells and seeded in ultra-low attachment round-bottom 96-well plates (Corning, 7007), containing human embryonic stem cell (hESC) medium supplemented with 4 ng/ml bFGF (PeproTech, AF-100-18B-50) and 50 µM Rho-associated protein kinase (ROCK) inhibitor (Merck Millipore, SCM-075). On day 3, the medium was replaced with fresh hESC medium. From day 5, the embryoid bodies were transferred to ultra-low attachment 24-well plates (Corning, 3473) in Neural Induction Medium. On Day 11, organoids were embedded into droplets of Matrigel (Corning, 354234) and transferred into 6-well plates in another two-day Neural Induction Medium. Differentiation Medium without vitamin A was fed from day 13 to day 19. From day 20 on, organoids were cultured in Differentiation Medium with vitamin A. Agitation was employed from day 18 at 70 RPM, and the medium was changed every 2-3 days.

## Transplantation of GBM cells for PDA generation
For injection of retrovirally transduced primary GBM cells into HBOs, $5 \times 10^4$ cells were cultured as a single cell suspension overnight. Newly formed spheres were resuspended in 1 µl of Differentiation Medium with vitamin A, loaded into a NanoFil syringe, and injected into the core of 2–3-month-old organoids under a dissection microscope. Tumor-bearing organoids were maintained in Differentiation Medium with vitamin A on an orbital shaker (70 RPM) for 15 days at 5% $CO_2$ and 37 °C.

## Orthotopic injection of GBM cells for PDX generation
For orthotopic injection of retrovirally transduced (or untransduced control) primary GBM cells into the mouse brain, $5 \times 10^5$ cells were cultured as a single cell suspension overnight. Newly formed spheres were resuspended in 2 µl of Matrigel, loaded into a NanoFil syringe, and stereotactically injected into the striatum (2.0 mm lateral to the bregma at a depth of 3.0 mm) of 8-10 week-old Fox Chase SCID-Beige mice under general anesthesia and perioperative pain management.

Tumor growth was longitudinally monitored by magnetic resonance imaging (MRI) at the Small Animal Imaging Center of the German Cancer Research Center. In addition, injected mice were monitored daily for disease symptoms. Animals with signs of distress were monitored and evaluated for the severity of symptoms and the need for euthanasia using a standardized termination score sheet, outlined in the approved study protocol. The score sheet included categories such as behavioral and neurological signs, as well as body weight, with numerical scores assigned to different levels of severity. Upon reaching these criteria, mice were sacrificed, and brains were collected after transcardial perfusion.

For perfusion, mice were anesthetized by intraperitoneal injection of perfusion solution. After opening the thoracic cavity, exposing the heart, transcardial perfusion was carried out with ice-cold HBSS.

### Flow cytometry analysis and fluorescence-activated cell sorting (FACS)

For isolation of TCF/Lef-EGFP reporter healthy v-SVZ-lineage, v-SVZ, rostral migratory stream (RMS), and olfactory bulb (OB) were dissected and single cell suspensions ($n = 5$ mice) were prepared using Neural Tissue Dissociation Kit (P) and gentleMACS Octo Dissociator with Heaters (Miltenyi) according to manufacturer's instructions. To allow discrimination of different populations, single cell suspensions were stained with the following antibodies conjugated with appropriate fluorophores as previously reported in ref. 15: anti-CD45, anti-Ter119, anti-O4, anti-Glast, anti-CD133 (Prom1) and anti-PSANcam, including proper unstained, single stained, and fluorescent minus one (FMO) controls (see Supplementary Data 7 for the list of antibodies used). After excluding dead cells, doublets, and CD45+/Ter119+/O4+ non-lineage populations; NSCs in the v-SVZ were gated based on Glast+vity and CD133+vity; early neuroblasts (ENBs) in the v-SVZ, as well as late neuroblasts (LNBs) in the OBs, were gated based on Glast-vity, CD133-vity, and PSANcam+vity. Each population was index-sorted into 384-well plates (Eppendorf Lobind) as single cells. Microplates containing cell lysates were briefly centrifuged, snap frozen on dry ice, and stored at −80 °C until processed for Smart-seq3.

For isolation of the xenografted human GBM cells, tumor-bearing mouse brains (5−7 mpi; $n = 8$ mice in total; $n = 4$ control; $n = 3$ SFRP1-OE; $n = 1$ Notum-OE) were dissected, and a single cell suspension was prepared using the Brain Tumor Dissociation Kit (P) and gentleMACS Octo Dissociator with Heaters (Miltenyi) according to the manufacturer´s instructions. Transduction marker mCherry was used to discriminate transduced human GBM cells, while a human-specific-MHCI antibody was employed to discriminate untransduced human GBM cells. After excluding dead cells and doublets, the mCherry+ (EGFP+ and EGFP−) cell population was index-sorted into 384-well plates (Eppendorf Lobind) as single cells. Microplates containing cell lysates were briefly centrifuged, snap frozen on dry ice, and stored at −80 °C until processed for Smart-seq3.

For isolation of HBOs and allografted human GBM cells, tumor-bearing organoids were dissociated using the Brain Tumor Dissociation Kit (P) and gentleMACS Octo Dissociator with Heaters (Miltenyi) according to the manufacturer's instructions. Prior to multiplexing ($n = 8$ HBOs), single cell suspensions were labeled with TotalSeqA anti-human hashtag oligonucleotide (HTO; Supplementary Data 7) antibodies (Biolegend, 1:50) to allow downstream de-multiplexing in silico as previously described in ref. 88. After excluding dead cells and doublets, mCherry+, as well as mCherry- cell populations, were sorted as bulk populations into 1.5 ml tubes (Eppendorf) and immediately processed for scRNA-seq using the 10× Chromium 3' sequencing platform.

Sytox Blue (Life Technologies, 1:1000) was used in all experiments as a dead cell indicator. Index sorting of single cells, as well as sorting of bulk samples, was carried out using a 100 micron nozzle at a BD FACSAria II or BD FACSAria Fusion at the Flow Cytometry Core Facility at the German Cancer Research Center.

### Single-cell library preparation with Smart-seq2/3

scRNA-seq libraries from v-SVZ-lineage populations were prepared using the Smart-seq3 protocol as previously described (Hagemann-Jensen, Nature Biotechnology, 2020) with minor modifications. The protocol was automated and miniaturized by incorporating liquid handling platforms, including Mosquito HV (SPTLabtech), Mantix (Formulatrix), and Viaflo 384 (Integra). Briefly, plates were incubated at 72 °C for 10 min to facilitate lysis and denaturation of secondary structures in the RNA. Lysed cells were subjected to reverse transcription in 2 μl using Maxima H-minus reverse transcriptase (Thermo Scientific), an oligo(dT) primer, and a template-switching oligonucleotide encompassing an 8-bp unique molecular identifier (IDT; see Supplementary Data 7 for all oligonucleotide sequences used). In a randomly selected set of wells, we included ERCC Spike-ins (Ambion) at a 1:2,500,000 dilution. Full-length cDNAs were amplified for 22 cycles of PCR using KAPA HiFi DNA polymerase (KAPA Biosystems). cDNA samples were purified with Ampure XP beads at a 1:0.8 ratio, and cDNA quality in randomly selected 10 wells/plate was assessed on a High Sensitivity Bioanalyzer chip (Agilent). cDNA concentrations were quantified using the Quant-it PicoGreen dsDNA Assay kit (Thermo Scientific) and the Synergy LX multi-mode microplate reader (Biotek). cDNAs were normalized to 250−500 pg μl$^{-1}$. The 100−200 pg of cDNA per sample was used for tagmentation in 1.2 μl using the Illumina XT DNA sample preparation kit. Libraries were finally amplified for 11 cycles of PCR in 4 μl using custom-designed Nextera index primers containing 8-bp index barcode sequences with a minimal Levenshtein distance of 4 as previously published in ref. 89. Samples were purified with Ampure XP beads at a 1:0.8 ratio, and DNA quality in randomly selected 10 wells/plate was assessed on a High Sensitivity Bioanalyzer chip (Agilent). Libraries were quantified as mentioned above, and normalized libraries were equimolarly pooled and purified one last time at a 1:0.8 ratio. Prior to sequencing, final library concentration was determined using Qubit dsDNA High Sensitivity Assay kit (Thermo Scientific) and Qubit fluorometer (Invitrogen), and the average fragment size was calculated on a High Sensitivity Bioanalyzer chip (Agilent).

scRNA-seq libraries from PDXs were prepared initially prepared using Smart-seq2[90] ($n = 4$) and later using Smart-seq3 protocol ($n = 6$) as outlined above. Libraries with 1% spiked-in PhiX control (Illumina) were sequenced at the 75-bp paired end on a high-output flow cell using an Illumina NextSeq550 instrument at a sequencing depth of ~1 M reads per cell at the sequencing open lab of the German Cancer Research Center.

### Single-cell library preparation by 10× Chromium 3´ sequencing

scRNA-seq libraries from HBOs, as well as patient-derived allografts, were prepared using Chromium Next GEM Automated Single Cell 3´ Library and Gel Bead Kit v3.1 (10× Genomics) according to the manufacturer's instructions at the single-cell open lab at the DKFZ. In parallel, HTO libraries were prepared as previously described in ref. 88. mRNA libraries with 2% spiked-in PhiX control (Illumina) were sequenced at the 100-bp paired end on a P3 flow cell using an Illumina NextSeq2000 instrument at a sequencing depth of ~80 K reads per cell. HTO libraries with 40% spiked-in PhiX control were sequenced at the 75-bp paired end on a mid-output flow cell using an Illumina NextSeq550 instrument at a sequencing depth of ~2 K reads per cell at the sequencing open lab of the German Cancer Research Center.

### Bulk whole-genome bisulfite sequencing (WGBS)

Tumor-bearing mouse brains (5−7 mpi) were additionally perfused with 4% PFA and post-fixed in 4% PFA overnight at 4 °C. The 50 μm coronal sections were prepared using a Vibratome VT1200S (Leica). Samples were stored in 0.01% NaN$_3$ in PBS at 4 °C until processing. The genomic DNA (gDNA) from three whole-brain sections per group was isolated by QiaAmp DNA microkit (Qiagen 1048145), and the

concentrations were measured by Qubit DNA HS kit (Thermo Scientific). The protocol for bisulfite conversion and library preparation was adapted based on the genomic part of the scNMT-seq protocol[91] with the following modifications. The gDNA input was adjusted to be 20 ng per sample. The number of first-strand purification cycles was decreased to four runs, and the library amplification reaction was performed in ten cycles. The library concentrations were measured by the Qubit DNA HS kit (Thermo Scientific), and the fragment distribution was analyzed by the Bioanalyzer DNA HS kit (Agilent). Before sequencing, libraries were pooled equimolarly. The final pool with 15% spiked-in PhiX control was sequenced at the 150 bp paired end on a P2 flowcell using an Illumina NextSeq 2000 at a sequencing depth of 40 M reads per sample at the sequencing open lab of the German Cancer Research Center.

### Immunofluorescent staining and microscopy

To visualize healthy v-SVZ-lineage, mouse brains were additionally perfused with 4% PFA and post-fixed in 4% PFA overnight at 4 °C. 50 μm coronal or sagittal sections were prepared using a Vibratome VT1200S (Leica). Samples were stored in 0.01% NaN₃ in PBS at 4 °C until staining. For staining, sections were incubated in blocking solution for 1 h at room temperature to block unspecific binding sites. Subsequently, sections were stained with anti-GFP, anti-GFAP, anti-Sox2, anti-S100β, anti-DCX, and anti-NeuN antibodies (Supplementary Data 7) in blocking buffer overnight at 4 °C on a rotator. After washing the sections three times 15 min with 0.25% Triton-X 100 in PBS, secondary antibody staining was carried out using appropriate Alexa-fluorophore antibodies (Supplementary Data 7) diluted in blocking buffer for 1 h at room temperature. DAPI was added to the antibody cocktail for nuclear staining. Furthermore, sections were washed three times 15 min with 0.25% Triton-X 100 in PBS. After mounting with Fluoromount-G, sections were allowed to dry in the dark at room temperature for 30 min. Samples were stored in the dark at 4 °C until imaging.

To visualize PDX tumors, mouse brains were prepared, processed, and imaged as outlined above. To visualize PDA tumors, HBOs were washed and fixed with 4% PFA for 1 h at room temperature, 10 dpi. Both PDA and PDX were stained with the following primary antibodies: anti-RFP, anti-GFP, and anti-human-specific Nestin prior to imaging.

Confocal images and tile scans were acquired in a Leica TCS SP5 or SP8 confocal microscope at the Light Microscopy Facility of the German Cancer Research Center. ImageJ was used for processing, and images were only adjusted for brightness and contrast.

### Spatially resolved transcriptomics by molecular cartography (MC; Resolve Biosciences)

For MC experiments, tumor-bearing mice were additionally perfused with 4% PFA and post-fixed with 4% PFA overnight at 4 °C. Samples were incubated in 15% and subsequently 30% sucrose solution overnight, each at 4 °C, respectively. OCT-embedded specimens were cryosectioned into 10 μm-thick coronal sections onto MC slides. Permeabilization, hybridization, and automated fluorescent microscopy imaging were performed according to the manufacturer's instructions for fix-frozen samples with minor modifications. Briefly, the slides were thawed at room temperature and dried at 37 °C in a thermal cycler with an MC slide holder. Sticky wells were attached to the MC slide to create the MC observation chamber. The sections underwent various treatments, excluding post-fixation but including permeabilization, rehydration, and application of TrueBlack autofluorescence quencher, which was diluted according to the instructions provided by Biotium. Following thorough washing and priming, specific probes designed by Resolve Biosciences' proprietary algorithm for our transcripts of interest (Sequences not listed here) were hybridized overnight at 37 °C. The hybridized sections were washed again, and the MC observation chamber was placed in the MC machine for eight automated cycles of coloring and imaging to determine the transcript localization of our panel of 98 transcripts in the tissue. Regions of interest (ROIs) were selected for each section (Control vs sFRP1) based on a brightfield overview scan. To aid the ROI selection process, we additionally scanned consecutive sections of the MC sections following immunofluorescent staining with tumor-specific markers (Supplementary Data 7). In the final imaging round, the nuclei were stained with DAPI to create a reference image for nuclei segmentation. After the run, the MC software registered the raw images, assigned transcripts to detected combinatorial color codes, and combined individual tiles to create ROI panoramas. The outputs consisted of text files containing the 3D coordinates of the transcripts and maximum projections of DAPI images for each ROI.

In order to complement the MC data with a panel of relevant proteins, processed sections were used to perform immunofluorescent labeling (as outlined above) of mCherry, EGFP, and were counter-stained for DAPI. The same ROIs were tile-scanned using Leica SP5 and SP8 confocal microscopes at the Light Microscopy Facility of the German Cancer Research Center, and images from respective RNA-protein modalities were registered using the common DAPI staining in each panorama.

### Data analysis

**Derivation of integrated adult WT SVZ NSC lineage and pseudotime.** Our reference lineage includes adult WT v-SVZ NSCs and progeny from three studies[15,19,37], supplemented with data from another study[92] for comparison. All datasets were reduced to 16,598 human-mouse 1:1 orthologs using Ensembl 102[93], with human gene names representing mouse counterparts. Counts, excluding the supplemental study, were integrated using the Seurat integration pipeline. The main NSC lineage was extracted via DBSCAN with eps = 0.65 on UMAP coordinates and integrated with the supplemental data to identify neuronal clusters. DBSCAN was used again to identify the main component, resulting in our SVZ NSC reference lineage. Cell-cycle scoring was carried out following the scanpy '180209_cell_cycle' example notebook with the supplied genes. Filtering for 1:1 orthologs, we scored 51 G2M- and 42 S-phase genes. We set a G2M-score cutoff at 0.1 to determine cycling activity. To generate an NSC lineage pseudotime, the PC2-lowest cell was chosen as the root for a diffusion map computed over PC1-3, and a branchless diffusion pseudotime[94] was computed over two diffusion components.

**SVZ QAD gene set and QAD state assignment across datasets**
We identified QAD cell types through their frequency in pseudotime by fitting a Gaussian KDE over the noncycling NSC lineage pseudotime and finding density peaks using argrelextrema from scipy.stats[95]. Pseudotime 0−0.282 denotes Quiescence (0−0.141 as dormant astrocytic stages), 0.282−0.676 denotes Activation, and >0.676 denotes Differentiation.

To capture QAD cell states across datasets, genes were prefiltered by global expression and per QAD state. Recursive feature elimination with a RandomForestClassifier on a 3-fold cross-validated 70/30 split reduced the feature set to 3333 genes. Then, 500 permutations of selecting 500 genes and training a RandomForestClassifier were performed, using the kneedle algorithm[96] to select pseudotime-predictive genes from ranked feature importances. Ribosomal genes were eliminated, and the top 100 of each state-enriched genes were selected, resulting in 242 genes assigned to QAD states based on maximum expression. GO-term enrichment was carried out using the enrichr class from gseapy[97] with expressed genes as background.

Human brain development scRNA-seq datasets were retrieved from the meta-atlas compiled by ref. 38 and reduced to 1:1 human-mouse orthologs. Samples with <300 cells were excluded, and QAD-stages were determined by maximum AUCell score, as at this point,

ptalign is not yet described. NSC lineage cell types were identified, and the mean proportion of QAD-stages per time point was calculated. QAD-stages were further compared to 'Type.v2' cell type labels.

For in vitro human NSC lineages, four PDA HBOs were sequenced by 10X, distributing tumor and healthy FACS-sorted populations by lane. Fastq files were processed with Cellranger v6.0.0. Organoid-hashtags were counted by CITE-seq-count[98], assigning cells based on a 1.5-fold threshold of the most frequent to the second most frequent hashtag. Cells with >500 genes, >2000 UMIs, and <20% mitochondrial transcripts were retained. Healthy and malignant cells were separated using inferCNV[99] with 10X brain nuc-seq cells as a reference. inferCNV was run on genes with a mean lognorm expression value > 0.25. Cells clustered by their 10X lane, and a Gaussian mixture model distinguished healthy and malignant cells using CNV calls. Lane- and CNV-concordant cells were retained for downstream analysis. The human NSC lineage was annotated by integration, and DBSCAN-isolated healthy organoid cells were reclustered in a human NSC-lineage UMAP. We identified and excluded 847 cycling cells and derived a lineage diffusion pseudotime as above.

## ptalign for pseudotime alignment

The ptalign program requires a query counts table, a reference counts table, a reference pseudotime mapping, and a set of trajectory genes. Query cells are placed along the reference pseudotime based on their gene expression correlation with trajectory genes. Reference cells are binned along pseudotime, mean expression per bin is calculated, and Pearson correlations are computed between query cells and pseudotime-binned references to represent a pseudotime similarity profile. Correlation scores from all cells are summarized in a correlation matrix, which is normalized and scaled. ptalign uses the supplied reference counts and reference pseudotime to compute a reference-reference correlation matrix and train a basic 3-layer multi-layer perceptron (MLP) network to predict the known reference pseudotime from the similarity profiles in the supplied matrix. A 5-fold cross-validated grid search optimizes MLP hyperparameters. Query cell pseudotimes are predicted by the trained network from input similarity profiles, with cell state inference based on reference pseudotime cutoffs. The quality of an assigned pseudotime is determined via DTW of the pseudotime-binned reference and an equally binned query. Counts are log-normalized, and Pearson correlations are computed for all combinations of reference and query bins. A matrix traceback is computed by dynamic programming to determine a path of maximal correlation through the DTW matrix. ptalign performance is estimated by comparing the length and average correlation along the DTW traceback. These metrics are compared to reference-query matrices computed according to equally sized and equivalently expressed permuted gene sets to determine an empirical P-value.

## Smart-seq3 datasets of GBM and v-SVZ and aligned pseudotimes

For GBMs, six 384-well plates of sorted T6 PDX tumor cells were prepared, two with Smart-seq3 and four with Smart-seq2. Reads were trimmed and mapped to the Ensembl 102 human and mouse genomes using STAR v2.5.3a. Cells were assigned to human, mouse, or QC-fail based on alignment rate and a human read ratio >1.3. The 1,176 QC-passing T6 Wnt-reporter cells were integrated using the Seurat pipeline, and cycling cells were called as above. Pseudotime alignment was performed with default parameters, and QAD cell states were assigned. This process was repeated for T6 PDA cells from above. Wnt activity in PDX cells was determined by FACS sorting relative to a reporter-positive control by index sorting. For PDA, we sorted by Wnt activity into separate 10X lanes.

For the TCF-Lef reporter v-SVZ dataset, five 384-well plates of sorted mouse brain populations from SVZ or OB were prepared using the Smart-seq3 protocol. Reads were trimmed with trim_galore[100] and mapped to the mm10 mouse genome using STAR v2.5.3a[101], then processed with a custom HTSeq count[102] script. Cell counts were QC-processed, log-transformed, and the NSC lineage was isolated by DBSCAN in UMAP space. Cycling cells were removed to produce a non-cycling lineage pseudotime. Pseudotime alignment was performed with default parameters, and QAD cell states were assigned using pseudotime cutoffs. Non-cycling lineage diffusion pseudotime and ptalign pseudotimes were compared by Pearson correlation per replicate to benchmark ptalign. This process was repeated for the HBO NSC lineage and pseudotime described above. Wnt activity in index-sorted PDX cells and TCF-Lef reporter SVZ reference lineage cells was assessed by fluorescence readouts relative to a reporter-positive reference gate.

## Aligned pseudotimes for published external GBM datasets

We retrieved published primary GBM single-cell RNA-seq counts tables, listed in Supplementary Data 2. Datasets were processed into an annotated counts table and subset to human-mouse 1:1 orthologs. All 66 datasets were merged, and GBM cells with >30% mitochondrial reads or <631 UMIs (corresponding to the smallest cell in the v-SVZ reference) were excluded. Tumor samples with fewer than 500 cells were also excluded, leaving 55. Counts were log-normalized per sample. Malignant cells were identified using inferCNV with a healthy CNV reference of 300 cells from our healthy HBO dataset. CNV genes were determined by comparing expression levels in healthy HBOs, our v-SVZ dataset, and GBM samples. Nonmalignant cell types were identified in UMAP space, and microglia, immune cells, and oligodendrocytes were excluded as in ref. 4. Cells with a Pearson correlation <0.4 to each sample's mean CNV calls were excluded, as well as tumors with <400 malignant cells, following the method outlined in ref. 4. Cycling cells were excluded as described above, and pseudotime alignment and QAD assignment were carried out for each tumor sample individually.

## QAD states association with tumor clones by CNV

Clones were identified using hierarchical clustering on inferCNV-predicted CNV values. A linkage matrix was computed for each tumor, and the first three splits were used to define clones per tumor, reducing them to three or four putative clones, which were further screened for CNV differences. CNV events >1.15 were considered gains and <0.85 losses. Tumors with fewer than 100 CNV events per cell were excluded. The ratio of CNV events in daughter clones was computed, and tumors where clones differed by <10% were excluded. For each of the remaining clones, we took the ratio of the average number of CNV events per cell per clone and compared it to the value at the parent node. A $z$-score threshold of 0.5 identified clones with diverging CNV, and we assessed expression divergence in these clones by Pearson correlation.

## ptalign pseudotime coherence vs conventional trajectory algorithms

For each of CytoTrace[103], Monocle3[104], and Palantir[105], pseudotimes for the 51-GBM cohort samples were determined according to the basic example in the respective documentation. For Monocle3 and Palantir, which require a root cell, the lowest cell in the ptalign pseudotime was chosen. These pseudotimes were compared to the ptalign pseudotimes by pseudotime coherence. Per sample, a 10-component PCA was computed, and the cell's pairwise Euclidean distances were determined. As conventional pseudotimes always span [0–1], we used rank distances to represent pseudotime distance. These were compared to PCA distances by Pearson correlation, with the method-based differences in coherence assessed by a paired $t$-test.

## ptalign permutation benchmark in non-brain tissues

The ptalign permutation model was benchmarked against non-brain tissue tumors from ref. 48. Annotated malignant cells were isolated from

colon ('Data_Lee2020_Colorectal'), breast ('Data_Qian2020_Breast'), and lung ('Data_Kim2020_Lung') datasets. Samples with >500 cells were retained, cycling cells identified, and reduced to 1:1 human-mouse orthologs. ptalign was run per sample with default parameters for 100 permutations, with the *P*-values recorded.

### Reproducing Genes2Genes results on the IBF dataset

We reproduced the healthy-vs-disease comparison shown in Fig. 5 of the Genes2Genes publication[46]. The datasets used are available from the Genes2Genes zenodo repository: 10.5281/zenodo.11182400. Gene sets were subset to human-mouse 1:1 orthologs, and cycling cells were excluded as above. We fit a banchless diffusion pseudotime through the healthy UMAP embedding, as described above, then split this lineage into 3 coherent Leiden clusters for which we called predictive genes using the recursive approach described above, resulting in a 241-gene gene set. Then we ran ptalign with these genes and default parameters. The Genes2Genes authors determine that the diseased trajectory deviates in the late pseudotime, so we subset early, middle, and late pseudotime increments and compute the area under smoothed spline curves representing pseudotime gene expression for each pseudotime increment using the Simpson function from scipy.integrate. We compute the area difference by taking the area under the absolute difference between healthy and diseased expression curves, and dividing it by the maximum coordinates along both healthy and diseased splines. We computed this ratio separately for each pseudotime increment. In Genes2Genes, EMT is enriched among genes deviating in late pseudotime, so we calculated the slope of the area difference ratio across pseudotime increments to identify those genes where the expression deviation is increasing, and then conducted GSEA on genes ranked by this slope.

### GBM-derived QAD geneset and application on bulk datasets

We compared expression in GBM QAD stages with >20 cells to healthy v-SVZ QAD expression. Genes seen in >80% of tumors were filtered by mean QAD-maximum lognorm expression. We computed normalized ranks of each gene by expression strength per state per tumor, selecting genes with >60% overrepresented ranking in a single state. These genes were compared to their state expression in the v-SVZ reference, again selecting by 60% enrichment in a given QAD-stage. DEseq log fold-changes in QAD-pseudobulks were used to take up to 100 genes per state, resulting in a 271-gene signature comprising 100 Q-, 71 A-, and 100 D-genes. A 92-gene cycling gene set was additionally derived by DEseq LFC > 2.

We collected microarray and clinical data from the TCGA-GBM cohort and Wu et al[24]. TCGA clinical metadata was obtained from the TCGA portal (https://portal.gdc.cancer.gov/) and merged using case_id fields. Data from recurrent GBMs was excluded. We curated TCGA-GBM data from Agilent 244K Custom Gene Expression arrays (https://portal.gdc.cancer.gov/legacy-archive/search/f), creating a new annotation file by aligning probes to the non-redundant nucleotide database (https://ftp.ncbi.nlm.nih.gov/blast/db/FASTA/nt.gz) with annotation by mygene v1.8. Background correction and normalization by LOESS-smoothing were performed, retaining only successfully annotated probes. Probes were summarized into gene sets, and low-quality arrays were excluded based on zero-count probes. Wu et al.'s data were processed in parallel, and survival reported in months was transformed to days.

For survival analysis, RNA bulk and microarray counts for GBMs were scored for SVZ-QAD and GBM-QAD gene sets using the R GSVA package[106]. For survival analysis, we right-censored by patient death and considered survival intervals up to the 95th percentile. A Cox proportional hazards (lifelines[107]) regression was carried out on the continuous QAD- and cellcycle GSVA-scores individually when accounting for patient sex and age, then adjusting for multiple testing.

### Methylation datasets acquisition and RNA states prediction

We obtained TCGA-GBM samples processed by the EPIC450k array and matched them to their RNA counterparts. Methylation values from the Wu et al. cohort were retrieved from GSE121722. We filtered compiled probe data to 'cg' probes not on sex chromosomes, with MAF < 0.01. Variable methylation sites were computed by the kneedle algorithm, selecting a cutoff in the cumulative distribution of variance over mean, then used in PCA.

Intra-tumoral distances were compared in the Wu et al. tumor cohort, with pairwise Euclidean distances in methylation and RNA space computed for all tumor pairs, then compared by Pearson correlation. Elasticnet regression predicted GSVA QAD scores from methylation data in a grid-search paradigm. After a 66/33 train-test split, variable features were selected by taking the 5000 probes with the largest *z*-score in the methylation PC2-4 loadings, as well as 2000 probes with the highest correlation in QAD scores, resulting in a 10,259 probe feature set for training. Methylation betas at this probeset were normalized and scaled using the StandardScaler and used in a GridSearchCV with 5-fold cross-validation to optimize the mean squared error and determine the optimal l1_ratio, alpha, and fit_intercept parameters. The grid-search selected parameters were used to train models on the training data and evaluated on the test data.

### Dysregulated genes in GBM vs SVZ by EMD

We first select a representative 12,581 gene set expressed in both tumor and healthy contexts by requiring genes to be expressed above a threshold in 1% of cells per sample, for at least 10% of samples. Then we preprocess each sample and fit splines to represent pseudotime expression using the skfda package[108]. We define a grid over pseudotime at intervals of 0.05. For each sample, we group cells into bins along the pseudotime grid, then filter bins containing <20 cells. We define 5 basis splines over the range [0–1] with a smoothing parameter using L2 regularization over the second derivative. We fit splines per gene on the expression mean per bin per sample, clipping negative values at 0. We store the resulting spline coordinates, as well as area-normalized curves using the Simpson function from scipy.integrate. For each gene, we combine spline values from healthy samples and take their average, then area-normalize as above. We use the scipy.stats wasserstein distance (EMD) implementation to compute the distance of the averaged healthy curve to each of the area-normalized genes in tumor samples. Unexpressed genes receive a Wasserstein distance of 1. We further derive the v-SVZ Q–D expression bias from area-normalized splines by defining Q- and D-intervals in the pseudotime grid, then computing the ratio of Q-area over summed Q- and D-areas.

We identify recurrently dysregulated genes by their mean GBM-SVZ Wasserstein distance across GBMs. To avoid noise introduced by area-normalization, we focus on genes in the upper 50% of expression levels as inferred by spline area, with secondary filtering by expression levels in v-SVZ. Genes which are lost in >50% of tumors and ptalign genes are also excluded, leaving 5415 genes for consideration. We rank genes by mean EMD as indicated above, and use the kneedle algorithm to identify the inflection point. 169 genes fall beyond this point, comprising the dysregulated gene set. We use the GSEAPY Enrichr library to compute GO enrichments in this group, using non-dysregulated genes as background.

Genes from major signaling pathways are identified based on membership in the GO-BP 2023 ontology downloaded using gseapy. By retaining GO terms containing 'Wnt', 'BMP', 'Notch', or 'Smoothened' and not containing 'Negative', we pool genes associated with each pathway and count their occurrence by pathway. Genes are associated with the pathway in whose terms they most frequently appear, with a lower bound of 50% enrichment. As 'Wnt' contains the most terms in GO, we additionally exclude genes which occur only in

Wnt-associated terms but are seen <3 times overall, leaving 158 gene-pathway associations: 62 Wnt, 47 Notch, 31 BMP, and 18 Smoothened (SHH).

## Biomarkers and targets in Q- vs D-biased GBMs

GBMs are clustered by QAD- and cycling-proportions using ward linkage, and the first split is used to separate into two groups of QA- and AD-enriched tumors. We compute pairwise group EMDs similar to above, additionally accounting for within-group consistency of EMD values. Thus, spline values along the pseudotime grid are collated for all tumors within a group, then EMDs are computed for each group member relative to the group mean. Finally, group-mean EMDs between QA and AD tumors are computed. We then filter genes where the within-group EMD is >0.1, leaving 6852 genes for consideration.

For selected biomarkers STMN1 and SOX11, and dysregulated genes KLF4 and ZFP36, we assayed bulk RNA expression and methylation levels by group in our TCGA cohort (see above). For methylation, we used the PC2-3 embedding of variable sites as above, to identify tumors where methylation aligned with dominant RNA stages: deriving 6 Q-enriched and 8 D-enriched tumors corresponding to QA- and AD-groups, respectively. Then we polled methylation probes covering our genes of interest, and plotted the methylation beta per sample, highlighting those probes where the absolute methylation difference between groups was >0.15. For RNA expression, we used differences in Q and D GSVA scores in the bulk TCGA and Wu et al. cohort, as above. Tumors with an absolute score difference > 0.6 were assigned to QA- and AD-groups, comprising 29 and 41 tumors, respectively, enabling expression comparisons by group.

## SFRP1 and NOTUM overexpression datasets

Sorted cells from T6 SFRP1- and NOTUM-overexpressing PDXs were sequenced, trimmed, and mapped as above. Cycling cells were removed, and pseudotime alignment was performed for QAD assignments. SFRP1 samples underwent pseudobulk DEseq analysis, followed by GSEA for enrichment among expression differences. Expression differences in the single NOTUM-OE replicate were quantified using pseudobulk LFC to Wnt-reporter Smart-seq3 replicates. T6 SFRP1-OE and Wnt-reporter cells injected into HBOs were sequenced on the 10X platform. Libraries were processed, and inferCNV distinguished malignant and healthy cell populations. Pseudotime alignment was carried out for tumor cells from each organoid. Healthy cells from SFRP1 and Wnt-reporter conditions were isolated, and cell types were identified by marker expression in a Leiden clustering. Genotypic comparison of healthy cell type expression was performed by averaging expression between pseudobulk replicates per cell type.

## WGBS data processing and SVZ VMR readout

WGBS libraries were prepared in triplicate from PDX samples as in ref. 91. A combined human-mouse reference genome was generated[109] to map reads from mouse and human cells using Bismark v0.2033[110] with Bowtie2 v2.3.5.134[111]. Paired reads were trimmed by trim_galore, and single-end reads individually mapped to the combined human-mouse reference with Bismark, setting −q -N 1 -D 30 -R 3 -p 6 with --non_directional. PCR duplicates were removed using deduplicate_bismark, and mate-pairs recombined by merging deduplicated R1 and R2 BAMs with samtools merge. Finally, we ran the bismark_methylation_extractor tool with --bedGraph to quantify DNA methylation and counted CpG coverage using coverage2cytosine. CpG coverages were parsed, and mouse and human methylation fractions per CpG were determined separately. v-SVZ cell type VMRs from ref. 30 were mapped to the human genome using liftOver, allowing up to 20% deviation in coordinates. VMRs with a genotype methylation difference > 1 st.dev from the mean were considered differentially methylated.

## Spatial alignment and preprocessing

For every ROI, Resolve Biosciences provided both 3D transcript positions and identities and a 2D DAPI image, which first needed to be integrated with our 3D confocal DAPI images. Resolve captures data in a tiled manner within every ROI; since they still had some stitching issues, every tile was separately aligned to the corresponding confocal image along the xy axes, using a homography obtained by matching SIFT features between the 2D DAPI image and a max-projection of the 3D DAPI image. Different angles of the slide in both microscopes were corrected by finding the angle of the slide plane in both modalities and rotating the transcript positions accordingly. The plane was fitted to the center of mass of the DAPI intensity along the z-axis for the 3D image, and to the tiled mean z position for the transcripts. Finally, a further shift along the z-axis was added such that the share of transcripts within segmented nuclei was maximized, using the segmentation described below.

## Segmentation

The 3D DAPI images were segmented into nuclei with mesmer[112], using the default pretrained model in the nuclear mode. Our images have a resolution of 0.142 microns per pixel, but to stay closer to their training data, we used 0.3 microns per pixel as the parameter for the segmentation. Segmentation was performed separately for every z-layer of the image, and afterwards stitched across the z-axis with a modified version of the intersection over union-based approach used in ref. 113. Segmentation of the ROIs into different brain regions was performed by hand.

The transcripts were then segmented with Baysor[114]. Baysor first clusters the transcripts by their neighborhood; we chose to demand 21 clusters, as this was the lowest number where each major cell type had at least one clear corresponding transcript cluster. Afterwards, it segments the transcripts into cells and background noise. Where transcripts intersected directly with segmented nuclei, this was added as the prior segmentation with confidence 0.98. The scale parameter was set to 2 microns, and we chose a minimum of 3 transcripts per cell. We excluded the transcript mCherry since it appeared to contain a lot of technical artefacts.

The parameters for Baysor were chosen to encourage over-segmentation rather than under-segmentation to properly resolve dense areas like tumors. To finally arrive at real cells, we assigned the Baysor cells to their nearest segmented nuclei, taking care to only combine Baysor cells that correspond to the same cell type. Prior to this, we consolidated the Baysor clusters to get rid of cases where multiple clusters corresponded to the same cell type. The assignment was done iteratively, first assigning Baysor cells that have a clear overlap with nuclei, and afterwards assigning the remainder to their closest compatible nucleus, starting with the Baysor cells with the most transcripts. Transcripts that Baysor marked as noise, and Baysor cells that could not be assigned to a segmented nucleus, were dropped.

## Spatial data analysis

Cell types for mouse cells (>60% mouse transcripts) were obtained by associating Baysor clusters with cell types. To assign human cells, we marked the transcripts that are enriched in each state (e.g., Q+ transcripts are enriched in Q). Cells with less than 20% Q+ or A+, or 10% D+ transcripts, were marked as generic human. Cells with at least 0.6 times as much Q+ as A+ or D+ were assigned Q, with at least 0.6 times as much D+ as Q+ or A+ were assigned D, and the remainder were assigned A.

Neighborhood enrichment was calculated using squidpy[115], using a radius of 20 microns to construct the neighborhood graph. The permutations were performed on the respective brain regions, combining all relevant regions from different ROIs. Squidpy reports the enrichment

as the number of actual neighbors compared to the number of expected neighbors, normalized by the standard error of the expected number. This was multiplied by the square root of the number of cells in the permutation to obtain *z*-scores in terms of the standard deviation, since this is a measure of the effect size that does not scale with the overall number of cells that entered the computation.

## Reporting summary

Further information on research design is available in the Nature Portfolio Reporting Summary linked to this article.

## Data availability

Newly generated SmartSeq3 datasets of the mouse v-SVZ NSCs lineage are available from GEO with accession GSE240676, along with processed tumor PDX and PDA scRNA-seq and WGBS datasets. Raw tumor data subject to DPA are submitted to EGA with accession EGAS00001008155. Access to the data is controlled due to ethical and privacy considerations related to human-derived samples, in compliance with data protection regulations. Access will be granted to qualified researchers for non-commercial academic use following approval by the Data Access Committee. Researchers should submit a data access request via the EGA portal, which will be evaluated based on the proposed use. Requests will be reviewed and data made available as outlined in the DKFZ Data Transfer Agreement. Generated spatial transcriptomics data are submitted to Zenodo under accession code 8186500. Analyzed GBM scRNA-seq datasets which had been previously published were retrieved as follows: Richards et al.[8] (SCP503), Bhaduri et al.[5] (personal communication), Neftel et al.[4] (SCP393), Yuan et al.[50] (GSE103224), Couturier et al.[6] (GitHub), Wang L. et al.[51] (GSE138794), Jacob et al.[41] (GSE141946), Wang R. et al.[52] (GSE139448), and Chen et al.[53] (GSE141383). Previously published scRNA-seq data of the v-SVZ, which we re-analyzed in this study, were retrieved from the following sources: Cebrian-Silla et al.[92] (https://cells.ucsc.edu/?ds=svzneurogeniclineage), Kremer et al.[37] (GSE145172), Carvajal-Ibanez et al.[19] (GSE197217), and Kalamakis et al.[15] (GSE115626). Samples and metadata from the TGCA-GBM cohort are available through the NCI GDC Data Portal. Bulk GBM data from Wu et al.[24] was received in personal communication. Source data are provided with this paper. All data supporting the findings of this study are available from the corresponding author upon reasonable request. Source data are provided with this paper.

## Code availability

The ptalign software is available on GitHub (https://github.com/leoforster/ptalign). Jupyter notebooks detailing the analyses and figures in this study are deposited in Zenodo under accession code 14968415.

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

## Acknowledgements

We gratefully acknowledge the following DKFZ core facilities for their support and expert assistance: Center for Preclinical Research, Small Animal Imaging Center, Genomics and Proteomics (Microarray Unit and Sequencing Open Lab), Single Cell Open Lab, Flow Cytometry and Light Microscopy. We are thankful to the Division of Biostatistics for statistical support. We further thank the following members of the Division of Molecular Neurobiology for their technical support: Santiago Cerrizuela, Lukas Kremer, Hadil El-Sammak, Andres Sanz Morejon, Damian Carvajal Ibanez, Jooa Hooli, Katrin Volk, Kathrin Menzner, Mohammad Al-Shukairi, Heike Abendroth, Aylin Korkmaz, and all the members of our group and the members of the Somatic Evolution and Early Detection group for continuously providing critical feedback on the project, as well as the manuscript. 7TGC plasmid was a gift from Roel Nusse; RKRR-P2A plasmid was a gift from Edward Green. Figures 1a, b, c, d, f, 2a, f, 3a, f, g, h, 5a, 6a, 7a, and Supplementary Figs. 1a, f, 2a, 3a, 6a, 8c, i used assets from BioRender.com. OK was funded by the Helmholtz International Graduate School for Cancer Research. This work was supported by the German Cancer Research Center (DKFZ), European Research Council (ERC; REBUILDCNS and PEPS, no. 101071786), and the German Research Foundation (DFG; SFB1324). Last but most important, we are grateful to patients and their families for their generous donations to biomedical research.

## Author contributions

LCF developed a computational tool for analysis of ASA of tumors and conducted computational analysis; conceptualized, designed, and analyzed experiments and interpreted results; and wrote the

manuscript. OK developed experimental protocols, conceptualized, designed, and performed experiments and interpreted results, and wrote the manuscript. VW performed the analysis of spatial transcriptomics data; DPD, AT performed mathematical modeling; VA, MB, KCZ, NS, SK, JT, JB, ILB, and NGP contributed to the development of the experimental protocols and supported experiments. XM and HL provided HBOs; AS and CAO contributed to the analysis of TCGA data; PUL, KP, and CRW provided GBM samples and clinical data; SA and AG supervised computational analysis; AMC supervised mathematical modeling. AMV conceived and directed the project, conceptualized and designed experiments, interpreted results, and wrote the manuscript.

## Funding

## Competing interests
L.C.F., O.K., and A.M.V. have filed a patent application. The remaining authors declare no competing interests.

## Additional information

[1]Molecular Neurobiology, German Cancer Research Center (DKFZ), Heidelberg, Germany. [2]Combined Faculty of Mathematics, Engineering and Natural Sciences, University of Heidelberg, Heidelberg, Germany. [3]BioQuant Centre, University of Heidelberg, Heidelberg, Germany. [4]Institute of Mathematics, University of Heidelberg, Heidelberg, Germany. [5]Molecular Neurogenetics, German Cancer Research Center (DKFZ), Heidelberg, Germany. [6]Metabolic Crosstalk in Cancer, German Cancer Research Center (DKFZ) and German Cancer Consortium (DKTK), DKFZ Core Center Heidelberg, Heidelberg, Germany. [7]Department of Neurosciences, Montreal Neurological Institute-Hospital, McGill University, Montreal, QC, Canada. [8]Department of Neurosurgery, Ulm University Hospital, Ulm, Germany. [9]Computational and Molecular Prevention, German Cancer Research Center (DKFZ), Heidelberg, Germany. [10]DKFZ Hector Cancer Institute, University Medicine Mannheim, Heidelberg, Germany. [11]Medical Faculty Mannheim, University of Heidelberg, Heidelberg, Germany. [12]Interdisciplinary Center for Scientific Computing, University of Heidelberg, Heidelberg, Germany. [13]These authors contributed equally: Leo Carl Foerster, Oguzhan Kaya. ✉e-mail: a.martin-villalba@dkfz-heidelberg.de

