## [Transparent Peer Review file · Nature Communications]

Cross-species comparison reveals therapeutic vulnerabilities halting glioblastoma progression

Corresponding Author: Professor Ana Martin-Villalba

This manuscript has been previously reviewed at another journal. This document only contains information relating to versions considered at Nature Communications. Mentions of the other journal have been redacted.

Version 0:

Reviewer comments:

Reviewer #1

(Remarks to the Author)

This manuscript has developed well and presents a strong, clearly articulated study. The presentation is polished, the figures are thoughtfully designed and visually compelling, and the integration of computational methods and benchmarking is commendable. The authors effectively relate their findings to normal developmental processes and provide a valuable emphasis on the concept of cellular quiescence, which will be a helpful perspective for the field.

I have a few points that I believe must be addressed before publication:

Clarify and Justify the Concept of ASA:

Activation State Architecture (ASA) is central to the paper but not explicitly defined. Based on the current framing, I interpret ASA to represent tumor cells' combined differentiation and proliferative status. While the terminology is evocative, it's unclear whether it adds conceptual clarity beyond existing frameworks. The authors should clearly define ASA early in the manuscript and justify its conceptual value. If it simply refers to quantifying differentiation states, it may be preferable to use more direct language rather than relying on potentially ambiguous terminology.

Avoid Overinterpretation of Linear ODE Dynamics:

In Figure 3c, the observed equilibrium state is a known and expected outcome of linear ordinary differential equation (ODE) models, reflecting convergence toward a dominant eigenvector. Similar dynamics have been described in state transition models of glioblastoma. As such, the interpretation that this behavior implies an "early establishment of growth-supporting architectures" may be overstated. The authors should temper this interpretation and clarify that this is a mathematical property of the modeling approach, not necessarily a biological insight.

Address the Mesenchymal Cell State:

A significant limitation of the current analysis is the absence of a discussion regarding the mesenchymal cell state, which is frequently observed in GBM. The authors should address how their model accounts for or fails to capture this population. Ideally, they would comment on where mesenchymally transformed cells appear in their pseudotime trajectory or explain their absence. This discussion is critical for comprehensively understanding the tumor's cellular diversity.

With these revisions, I would be supportive of the manuscript's publication. The work represents a meaningful contribution to the field and, with the above clarifications, will be even more impactful.

(Remarks on code availability)

Reviewer #3

(Remarks to the Author)

I appreciate the additional data efforts and clarifications - I feel the manuscript has been sufficiently edited and addresses my

major concerns. I now endorse publication.

(Remarks on code availability)

I looked at but did not run the code. It is well documented enough that it should be possible to execute clearly.

REVIEWERS' COMMENTS

Original Reviewer #1 at [REDACTED] (Remarks to the Author):

This manuscript has developed well and presents a strong, clearly articulated study. The presentation is polished, the figures are thoughtfully designed and visually compelling, and the integration of computational methods and benchmarking is commendable. The authors effectively relate their findings to normal developmental processes and provide a valuable emphasis on the concept of cellular quiescence, which will be a helpful perspective for the field.

We thank the reviewer for their thoughtful and supportive comments. We appreciate the positive assessment of our study's methodology, presentation, and conclusions, and share their sentiments with regard to the positive impact our work will have on the field.

Presently, we address the remaining points made by the reviewer:

I have a few points that I believe must be addressed before publication:

Clarify and Justify the Concept of ASA:

Activation State Architecture (ASA) is central to the paper but not explicitly defined. Based on the current framing, I interpret ASA to represent tumor cells' combined differentiation and proliferative status. While the terminology is evocative, it's unclear whether it adds conceptual clarity beyond existing frameworks. The authors should clearly define ASA early in the manuscript and justify its conceptual value. If it simply refers to quantifying differentiation states, it may be preferable to use more direct language rather than relying on potentially ambiguous terminology.

The Activation State Architecture (ASA) is not simply a quantification of differentiation states, but it refers to the organized distribution of activation states among all tumor cells, forming a unique structural feature of each tumor. This architecture reflects an early equilibrium state from which key parameters governing tumor growth can be inferred. ASA serves as a prognostic marker, supports patient stratification, and informs personalized therapeutic interventions. These points are now emphasized in the text as outlined below.

Changes to the text (Introduction, p. 2):

To address these questions, we investigate Activation State Architectures (ASAs): unique structural features that represent the organized distribution of activation states in each tumor. These ASAs have the potential to serve as a prognostic marker, support patient stratification, and inform personalized therapeutic interventions.

Changes to the text (Discussion, p. 10):

Here, we leveraged the similarities between GBMs and adult NSCs to decode Activation State Architectures (ASAs) using ptalign, a computational method that aligns tumor cells within a reference pseudotime axis. We show that this architecture reflects an early equilibrium state from which key parameters governing tumor growth can be inferred. Together, ASAs serve as prognostic markers, support patient stratification, and inform personalized therapeutic interventions.

The reviewer continues:

Avoid Overinterpretation of Linear ODE Dynamics:

In Figure 3c, the observed equilibrium state is a known and expected outcome of linear ordinary differential equation (ODE) models, reflecting convergence toward a dominant eigenvector. Similar dynamics have been described in state transition models of glioblastoma. As such, the interpretation that this behavior implies an "early establishment of growth-supporting architectures" may be overstated. The authors should temper this interpretation and clarify that this is a mathematical property of the modeling approach, not necessarily a biological insight.

We thank the reviewer for pointing out this misleading formulation. The invariance of tumor architecture is indicated by published data of PDX and PDA tumors showing the same proportions of cellular states found in the original patient sample (e.g. Neftel et al., 2019, Jacob et al., 2020, and Peng et al., 2025). As such, the intrinsic properties of linear ODE models, as noted by the reviewer, are in agreement with experimental observations, justifying their use. The models allow to quantify the relationship between time to equilibrium and time to detection, showing an early establishment of tumor architecture. In addition, preliminary simulations using nonlinear ODE models that incorporate regulatory feedback—adapted from those we recently developed for the healthy v-SVZ (Danciu et al., 2025)— show similar behavior for exponentially growing tumors in terms of both capability of recapitulating different architectures and early establishment of QAD proportions. We have added two panels in Supplementary Fig. 2 to highlight the early adoption of primary ASAs in GBM organoids, in line with the model predictions, and adapted the main text to provide a better explanation:

Supplementary Fig. 2 | QAD stage inference across modalities and time

e Stacked barcharts depicting the proportion of QAD-stage cells in primary tumor tissue (Tumor) as well as 2, 8, and 24 weeks (wk) in culture. Data are from Jacob et al.

f Compositional distances between tumors from (e) and corresponding *in vitro* culture times indicate early establishment of ASA which remains temporally stable. For reference, horizontal dotted lines denote compositional differences from Fig. 1g, as indicated.

Changes to the text (p.6):

Though such dynamics can be expected from the class of linear ODEs applied here (see Supplementary Data 3), it is consistent with recent non-linear ODE models of the v-SVZ⁵⁹ as well as the early establishment and faithful recapitulation of cellular architectures observed in GBM organoids^{41,42} (Supplementary Fig. 2).

The reviewer continues:

Address the Mesenchymal Cell State:

A significant limitation of the current analysis is the absence of a discussion regarding the mesenchymal cell state, which is frequently observed in GBM. The authors should address how their model accounts for or fails to capture this population. Ideally, they would comment on where mesenchymally transformed cells appear in their pseudotime trajectory or explain their absence. This discussion is critical for comprehensively understanding the tumor's cellular diversity.

The reviewer raises a valid question regarding the inclusion of mesenchymal tumor cells in ptalign. We can reassure the reviewer that ptalign is able to account for mesenchymal tumor cells, and have generated an additional supplemental panel to (new Supplementary Fig. 6a) demonstrate this point. There, we assigned cells in our GBM cohort to Neftel metamodules by AUCCell and averaged pseudotime cell-densities to determine a 90% highest-density interval where the majority of each metamodule's cells can be found in pseudotime. This confirmed that MES-like tumor cells are found in quiescent states, but also confirmed the Q-bias of AC-like cells along with the D-bias of NPC-like cells. These points are reflected in the text as outlined below.

Supplementary Fig. 6 | Leveraging expression heterogeneity in GBMs to identify genes with consistent and divergent expression dynamics

a Placement of GBM metamodules from Neftel et al within ptalign pseudotime. Metamodule assignment by max AUCCell > 0.2. Bars denote 90% highest-density interval, from averaging cell density across indicated number of GBMs with >10% of metamodule-assigned cells.

Changes to the text (p. 5):

Existing GBM metamodules from Neftel et al⁴ are assigned to their expected QAD-stages (Supplementary Fig. 6a), including a distinct Q-bias for AC-like (astrocyte-like) GBM cells and D-bias among NPC-like cells. Further underscoring the advantage of the ptalign mapping, we find MES-like (mesenchymal) GBM cells split between Q- and A-stages in line with the reported link of quiescence to hypoxic and activation to wound-response signatures in the mesenchymal population^{51,55,56}. Thus, GBM ASAs decoded by ptalign offer functional insights into tumor organization, paving the way for individual patient stratification.

The reviewer continues:

With these revisions, I would be supportive of the manuscript's publication. The work represents a meaningful contribution to the field and, with the above clarifications, will be even more impactful.

We thank the reviewer for this assessment and hope that our stated revisions develop the impact and clarity of our manuscript.

Original Reviewer #3 at [REDACTED] (Remarks to the Author):

I appreciate the additional data efforts and clarifications - I feel the manuscript has been sufficiently edited and addresses my major concerns. I now endorse publication.

Reviewer #3 (Remarks on code availability):

I looked at but did not run the code. It is well documented enough that it should be possible to execute clearly.

We thank the reviewer for their positive assessment of our revised study, as well as for checking and assessing the code submitted with the manuscript.